# Lung structural cells are altered by influenza virus leading to rapid immune protection following re-challenge

Julie C. Worrell [1,6] ✉, Kerrie E. Hargrave[1], George E. Finney [1], Chris Hansell[1], John Cole[1], Jagtar Singh Nijjar[2], Fraser Morton [1], Marieke Pingen [1], Tom Purnell[1], Kathleen Mitchelson[1,6], Euan Brennan [1], Clíodhna M. Daly[1], Jay Allan[3], Georgios Ilia [3], Vanessa Herder [3,4], Claire Kennedy Dietrich[5], Yoana Doncheva[5], Nigel B. Jamieson [5], Massimo Palmarini [3,4] & Megan K. L. MacLeod [1] ✉

Lung structural cells form barriers against pathogens and trigger immune responses following infections. This leads to the recruitment of innate and adaptive immune cells some of which remain within the lung and contribute to enhanced pathogen control following subsequent infections. There is growing evidence that structural cells also display long-term changes following infection. Here we investigate long-term changes to mouse lung epithelial cells, fibroblasts, and endothelial cells following influenza virus infection finding that all three cell types maintain an imprint of the infection, particularly in genes linked to communication with T cells. MHCI and MHCII proteins continue to be expressed at higher levels in both differentiated epithelial cells and progenitor populations and several differentially expressed genes are downstream of the transcription factor, SpiB, a known orchestrator of antigen presentation. Lung epithelial cells from influenza-infected mice display functional changes, more rapidly controlling influenza virus than cells from naïve animals. This rapid antiviral response and increased expression of molecules required to communicate with T cells demonstrates sustained and enhanced functions following infection. These data suggest lung structural cells display characteristics of immune memory which could affect outcomes that are protective in the context of infection or pathogenic in chronic inflammatory disorders.

The lung is constantly exposed to microbes that can trigger immune responses. Respiratory viruses, including influenza A virus (IAV), are leading causes of disease worldwide[1,2]. The World Health Organization estimates annual epidemics result in ~3–5 million cases of severe illness and ~290,000–650,000 deaths.

Many different cell types are required to control the pathogens that enter the lung. These include structural cells, epithelial cells and stromal cells (fibroblast and endothelial cells); and immune cells that can be resident or recruited following the infection. The early responses to IAV by lung structural cells have been well

[1]School of Infection and Immunity, University of Glasgow, Glasgow, UK. [2]Weatherden Ltd, London, UK. [3]MRC-University of Glasgow Centre for Virus Research, Glasgow, UK. [4]Department of Viroscience, Erasmus Medical Centre, Rotterdam, The Netherlands. [5]School of Cancer Sciences, University of Glasgow, Wolfson Wohl Cancer Research Center, University of Glasgow, Glasgow, UK. [6]Present address: Conway Institute and School of Medicine, University College Dublin, Dublin, Ireland. ✉e-mail: Julie.worrell@ucd.ie; Megan.MacLeod@glasgow.ac.uk

characterized[3–8]. Epithelial cells are the primary cell type infected in vivo and are the cells most likely to produce new virus[9,10]. They, and nearby fibroblasts, trigger an early wave of inflammatory cytokines and chemokines that can help control the infection until dedicated immune cells migrate to the infection site[3–6]. Blood endothelial cells (BECs) play a key role in alerting circulating immune cells and recruiting them to the lung via chemokines[7,11].

IAV can be cleared within 7–10 days, but the consequences of the infection are more sustained. While epithelial cells and fibroblasts are involved in tissue repair, damage can persist for many months in mouse infection models[12–14]. Lung immune cells are also altered by infection: adaptive T and B cells can take up long-term residence in tissues as Tissue-resident memory (Trm) cells, and circulating monocytes can differentiate into alveolar macrophages. These cells can provide both pathogen-specific and non-specific protection to future challenges and are considered arms of adaptive and innate memory[15–22].

Recently, evidence is growing that structural cells can also display long-term changes to infections[23,24]. In the context of IAV, club epithelial cells that survive IAV infection can produce a rapid anti-viral response following a second challenge, which may contribute to tissue damage[25,26]. Other epithelial cells can also survive IAV infection by reducing communication with lung CD8 T cells[27]. However, an in-depth analysis of long-term changes by IAV to lung epithelial cells is lacking. Moreover, little is known about the long-term changes to other lung structural cells following influenza virus infection.

To address these questions, we used transcriptomic and flow cytometry analysis to investigate long-term changes to lung structural cells following primary infection and re-challenge of mice with IAV. Our data reveal that, in addition to epithelial cells, fibroblasts and BECs maintain an imprint of the IAV infection. Our analysis demonstrates that antigen processing and presentation is a shared pathway upregulated in all three cell types for at least 40 days following infection and implicates the Ets family transcription factor, SpiB[28], in regulating these changes.

More rapid control of IAV is expected in re-challenge infections, even when different IAV strains are used in a heterosubtypic infection to avoid neutralizing antibodies[18,21,29,30]. We find that lung epithelial cells rapidly upregulate the expression of molecules required for communication with T cells and present antigen to IAV-specific CD4 and CD8 T cells in vitro. Despite this ability to communicate with lung T cells, in vivo rapid virus control within the first two days of a re-infection is independent of T cells. Instead, our data show that epithelial cells from previously infected mice can more rapidly control IAV in vitro independently of immune cells. Together, our data suggest that lung structural cells maintain an imprint of IAV infection that can enhance their ability to communicate with T cells and, for epithelial cells, also bolsters their own immunity against subsequent infections.

## Results

### IAV infection causes prolonged upregulation of immune-related genes in lung structural cells

To advance our understanding of the long-term consequences of IAV infection on lung structural cells, we used fluorescence-activated cell sorting (FACS) to isolate three major cell types in the lung: epithelium, fibroblasts, and endothelium. We sorted the cells based on the expression of EpCAM1 (epithelial cells), CD140a (fibroblasts), and CD31 (BECs) from mice at day 10 (up to viral clearance) and 40 post-infection (long after viral clearance) and compared their gene expression to cells sorted from naive animals. The gating strategy, sort purities and Principal Component Analysis (PCA) are shown in SFig. 1A–E. PCA demonstrated substantial changes at day 10 post-infection for all cell types compared to naive controls. We identified 1016 differentially expressed genes (DEG) in epithelial cells (784 up, 232 down), 3128 DEG in fibroblasts (1772 up, 1356 down) and 1292 DEG in endothelial cells

(968 up, 324 down) between day 10 IAV and naive groups (Fig. 1A, B and Supplementary Data 1). Over Representation Analysis (ORA) detected enrichment in cell cycle in all three subsets and defense response to virus in epithelial cells and fibroblasts, SFig. 2A–C.

At day 40 post infection, all three cell types continued to show differential gene expression compared to cells from naive mice, although the number of DEG were reduced compared to day 10: epithelial cells: 144 up, 61 down; fibroblasts 159 up, 43 down; BECs: 58 up and 21 down (Fig. 1C, D). To validate the analysis, we isolated RNA from lung epithelial cells and fibroblasts in an independent experiment from naive mice and animals infected 30 days earlier and compared gene expression in cells from naive and infected animals by microarray (nCounter Nanostring mouse immunology panel with 29 custom genes, Supplementary Data 2). We found a consistent fold change between DEG identified in the RNAseq that overlapped with genes in the microarray (SFig. 3 and Supplementary Data 2).

Closer inspection of overlapping upregulated DEG from our RNAseq datasets at day 10 and 40 post-infection, revealed that several genes remained persistently elevated with 80, 131 and 43 DEG in epithelial cells, fibroblasts, and BECs, respectively (Fig. 1E and Supplementary Data 1). BECs had the lowest number of genes that remained upregulated, suggesting these cells may be more likely to return to homeostasis than epithelial cells or fibroblasts. ORA analysis of the DEG in the three cell types indicated sustained changes in pathways related to the innate and adaptive immune responses (SFig. 2D–F). Taken together, these data reveal sustained transcriptional alterations in immune-related genes in lung structural cells long after viral clearance.

We next performed a gene set enrichment network analysis on all structural cell types at both day 10 and 40 post-infection (Fig. 2A–C). This was performed as ORA can enrich for highly similar genes and may only represent a portion of the enriched biology. Network analysis revealed conserved upregulation of gene networks governing shared biological themes in all three cell types, including antigen processing and presentation at day 40 post-IAV infection. These findings indicate that IAV infection leads to long-term changes in the expression of immune-related genes in lung structural cells.

### Expression of *Spib* in airway epithelial cells found near immune cell clusters is dependent on viral replication

Analysis of the upstream regulators of the DEG identified at day 10 and 40 in epithelial cells and fibroblasts indicated a number of potential upstream regulators including ETS1, NFκB1 and CITED2 (SFig. 4A). These molecules are downstream of Interferon (IFN)γ, CD40L and IFNα which can be produced by multiple cell types during IAV infection including epithelial cells, T cells and NK cells[31]. Interestingly, both epithelial cells and fibroblasts had significantly elevated gene expression levels of the ETS transcription family member *Spib* at day 40 post-infection, and we confirmed increased expression by qPCR (SFig. 4B). SpiB has recently been shown to coordinate differentiation and chemokine production by splenic fibroblasts following lymphocytic choriomeningitis virus infection[32].

To investigate the potential contribution of SpiB to the transcriptional changes in lung structural cells following IAV infection, we compared the persistently upregulated DEG identified in our RNAseq analysis with genes identified as SpiB targets using a mouse ChIP-seq data set[33]. In epithelial cells, 25% of persistently upregulated DEG following IAV infection were SpiB target genes (20/80), in fibroblasts SpiB target genes (22/131) accounted for 17% of DEG (Fig. 3A and Supplementary Data 3). 27% of the SpiB targets induced by IAV infection were conserved between lung epithelial cells and fibroblasts (9/33): *H2-Q4, H2-Q5, H2-Q7, Psmb8, H2-K1, H2-Q6, SpiB, H2-Ab1, Psmb9* (Fig. 3B). These data indicate that structural cells share an imprint of previous infection, and SpiB may be a marker of this infection experience.

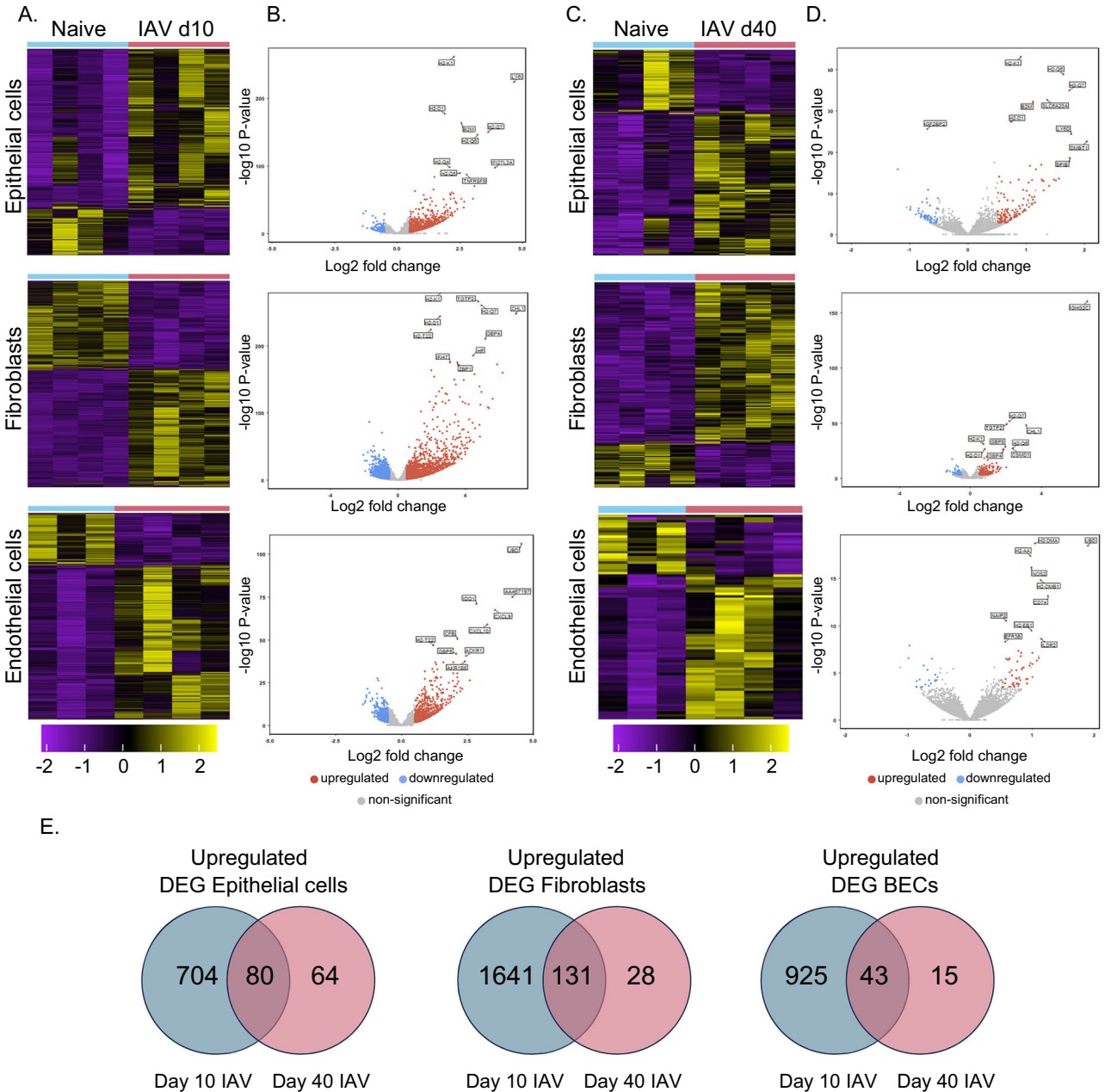

**Fig. 1 | IAV infection induces conserved transcriptional changes at day 10 and 40 post infection.** RNA sequencing was performed on sorted lung epithelial cells, fibroblasts, and blood endothelial cells (BEC) from naive and C57BL/6 mice infected with influenza A virus (IAV) 10 (**A**, **B**) or 40 (**C**, **D**) days previously. **A**, **C** Differentially expressed genes (DEG) with a twofold change and a False Discovery Rate $q$ value < 0.05 in lung structural cells were considered statistically significant. 7–8 mice were combined to make four samples per timepoint. **B**, **D** Volcano plots show the relationship between the fold change and the associated $p$ value with the top 10 DEG by fold change. **E** Venn diagrams showing overlap between differentially expressed upregulated genes at day 10 and 40 post IAV infection. Source data in **E** are provided as a Source Data file.

To determine the location of transcriptionally altered epithelial cells, we performed an RNAscope on paraffin-embedded lung tissue from naive animals or mice infected with IAV for 30–40 days previously. We found clusters of B220 + B cells that were SpiB+ cells in the parenchyma, as described by ourselves and others (SFig. 5)[17,34–38]. *Spib*+ epithelial cells were rare in the airway epithelium of naive animals but were increased in *Spib*+ structural cells in infected animals, and were only present in areas of the lung close to clusters of immune cells (Fig. 3C).

This link in location between infection-altered epithelial cells and immune clusters could indicate sites of viral replication during the initial infection. To ask if viral replication was required for *Spib* expression, we infected mice with a modified IAV virus, S-FLU, which, due to a mutation in the vRNA encoding HA, can only undergo one round of replication in host cells[39]. Despite this reduced replication, S-FLU does induce systemic and local adaptive immune responses[39,40]. *SpiB* expression was not detected in lung epithelial cells in animals infected with S-FLU 30 days previously (Fig. 3D). In addition, these mice did not display overt clinical signs of infection, with no reduction in total body weight following infection with S-FLU as previously reported[39] (SFig. 6A). At 30 days post infection lungs from mice infected with S-FLU had reduced levels of inflammatory infiltrate

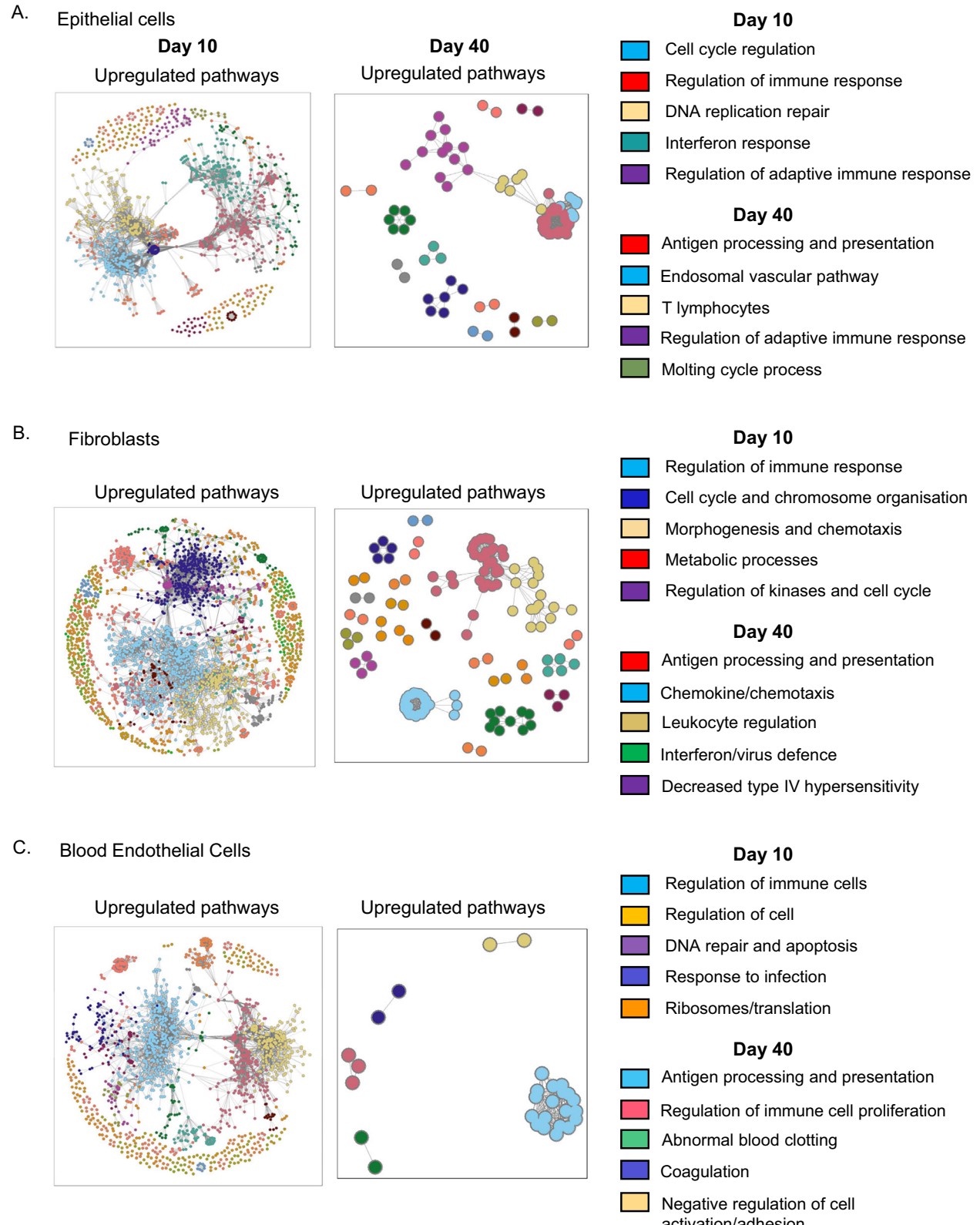

**Fig. 2 | Network analysis shows conserved upregulation of genes involved in antigen processing and presentation in lung structural cells at day 40 post-IAV infection.** Networks of all significantly enriched gene sets in epithelial cells **A**, fibroblasts **B**, and blood endothelial cells **C**, highlighting significance and gene set interconnectivity. Networks are given for enrichment amongst all significantly differentially expressed genes (*p*.adjusted <0.05, absolute log2 fold >0.5) between day 10 and naive, and day 40 and naive. Enrichment analysis was performed using Hyper-geometric Gene Set Enrichment on the gene set databases STRING11.5. For each network, nodes represent gene sets and edges represent two gene sets with a Szymkiewicz-Simpson coefficient of at least 0.5. Colors are defined in the Figure.

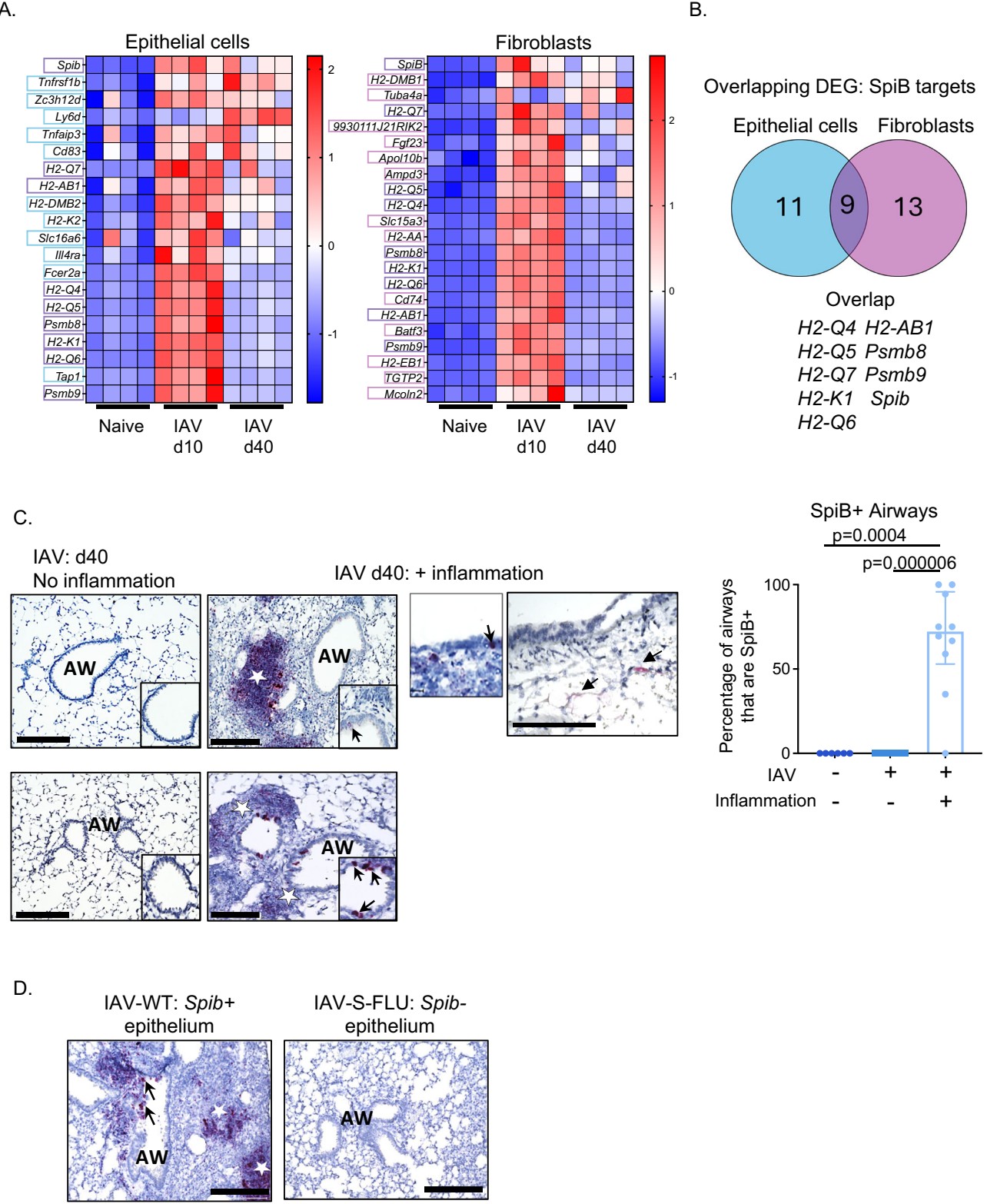

A. Epithelial cells / Fibroblasts heatmaps with Naive, IAV d10, IAV d40 conditions.

B. Overlapping DEG: SpiB targets. Venn diagram: Epithelial cells 11, overlap 9, Fibroblasts 13. Overlap: H2-Q4, H2-AB1, H2-Q5, Psmb8, H2-Q7, Psmb9, H2-K1, Spib, H2-Q6.

C. IAV: d40 No inflammation; IAV d40: + inflammation. SpiB+ Airways. p=0.0004, p=0.000006. Percentage of airways that are SpiB+. IAV − + +; Inflammation − − +.

D. IAV-WT: *Spib*+ epithelium; IAV-S-FLU: *Spib*- epithelium.

compared to those infected with the WT-IAV strain (SFig. 6B) shown by H&E staining (SFig. 6C). Viral replication was also necessary for the formation of immune cell clusters. These mice lacked B220+ and CD4+ cell-containing clusters, compared to IAV-X31-infected mice (SFig. 6D). Taken together, our findings indicate that viral replication is necessary for *Spib* expression by lung epithelial cells.

## IAV infection leads to increased MHCII expression by multiple epithelial cell subsets

The lung contains multiple subsets of epithelial cells that are specialized for different roles and that respond distinctly to IAV infection[3,26,41]. We used multi-parameter flow cytometry to investigate the consequences of IAV on lung epithelial cell subsets. We

**Fig. 3 | *Spib* is upregulated in lung epithelial cells adjacent to areas of inflammation, and this is dependent on viral replication.** Upregulated genes at both day 10 and 40 post-infection from the experiments described in Fig. 1 were compared with target genes of the transcription factor, SpiB, identified using a publicly available ChIPseq dataset (Solomon et al.[33]) **A** Heatmaps show SpiB target genes in epithelial cells and fibroblasts, respectively. Expression across each gene (row) is scaled, with a Z-score computed for all Differentially Expressed Genes (DEG). Targets specific to epithelial cells (blue boxes), targets specific to fibroblasts (pink boxes) and those that overlap between cell types (purple boxes). Samples with relatively high expression of a given gene are marked in red, and samples with relatively low expression are marked in blue. **B** Venn diagram shows overlap between SpiB target genes and DEG in lung epithelial cells (blue) and fibroblasts (pink) post IAV infection. **C** Lungs were taken from naive or C57BL/6 mice infected with influenza A virus (IAV) 30–40 days previously, with representative images

from mice from two independent experiments shown. Areas of infected lungs with no inflammation have *Spib* negative airways, while *Spib* (black arrows) is localized in areas of inflammation (dark red color, indicated by white star) within the lung and in airways; adjacent images showing areas of no inflammation and inflammation show lungs from the same mouse. Data are from 4 to 6 mice per experimental group, combined from two independent experiments. Each symbol represents a lobe from a mouse, with each mouse only represented once in each column, and bars show the median with the interquartile range, as data are not normally distributed. Data analyzed using Kruskal–Wallis with Dunn's post hoc comparison test. **D** C57BL/6 mice were infected intranasally with IAV (either wildtype (WT) or single cycle (S-FLU), Airways (Aw), blood vessels (Bv), inflammatory foci/immune cell clusters (labeled with white stars) and *Spib* positive airway epithelial cells (black arrows). All scale bars are 200 μm. Source data in **A** and **C** are provided as a Source Data file.

incorporated exclusion markers (CD45, CD31, CD140a) for immune, endothelial and stromal cells and gated based on a pan-epithelial marker EpCAM (CD326). Our gating strategy was further refined using additional surface markers (e.g., Sca1, Pdpn, MHCII, and CD24) to distinguish airway (club, ciliated, progenitor) and ATI and II epithelial subtypes. We used CD24 and Sca1 to identify distinct airway epithelial cell populations based on investigations by McQualter et al.[42], Chen et al.[43], and Lee et al.[44]. These groups identified multiple progenitor populations in the upper airway, CD24−Sca1+ [42,43] and CD24+Sca1− [42]. We also gated on CD24−Sca1− to identify alveolar epithelial cells, an approach consistent with Liang et al.[45], who used this strategy to identify alveolar epithelial cells. ATI epithelial cells were distinguished from ATII based on expression of cell surface marker Pdpn and MHCII[44,46].

We coupled this with a novel SpiB reporter mouse to explore further the potential role for SpiB in epithelial cell function. In these transgenic (Tg) animals, the sequence for a 2 A peptide (T2A[47]) and the fluorescent molecule, mCherry, were inserted into the *Spib* locus. As expected B cells[48] and plasmacytoid DCs[49,50] were positive for the reporter (SFig. 7A). We also sorted SpiB-mCherry negative and positive EpCAM1+ cells from naive mice and examined the expression of *Spib* by semi-quantitative PCR (SFig. 7B). SpiB-mCherry+ cells expressed low levels of *Spib* but higher levels than SpiB-mCherry-negative cells. This low expression level likely explains why we do not detect *Spib*+ cells in naive lungs by RNAscope. The SpiB-mCherry+ cells also expressed higher levels of SpiB target genes (*Psmb9, Tap1, Cd74, H2ab1, Tap1*[33]) and *Ifi47*, an Interferon-Stimulated Gene[51] (SFig. 7B).

In naive mice, SpiB was expressed by multiple epithelial cell types including those of the upper airway: ciliated (CD24hi Sca1+), club (CD24+ Sca1+), progenitor cells (CD24+ Sca1neg), Sca1+ progenitor cells (Sca1+ CD24neg), and alveolar epithelial cells (ATI: CD24neg Sca1neg Pdpn+ and ATII: CD24neg Sca1neg Pdpnneg MHCII+) from the distal lung (Fig. 4A); gating strategy is shown in SFig. 7C and based on previous studies[42–45]. In comparison to SpiB-negative EpCAM1+ cells, the SpiB+ population was more likely to contain ATII but fewer ATI, and CD24+ and Sca1+ progenitor cells (Fig. 4A).

As SpiB is known to regulate molecules involved in MHCII antigen processing and presentation[33] we examined the expression of MHCII on the epithelial cell populations, excluding ATI cells that are defined as MHCII negative[52]. For all the other populations, SpiB+ cells were more likely to be MHCII+ than SpiB negative populations, this included ATII cells that are predominately MHCII+ in which we found that SpiB+ cells were slightly, but significantly, more likely to express MHCII (Fig. 4B, SFig. 7D).

We investigated the impact of IAV infection on the number of SpiB-negative and positive epithelial cell populations at days 10 and 30 post-infection. IAV infection led to a decline of SpiB-negative and positive, CD24 progenitor cells and SpiB+ ATII cells at day 10 (Fig. 4C). In contrast, Sca1+ SpiB+ progenitors more than doubled at day 10

post-infection, while the numbers of SpiB-negative Sca1+ progenitors remained unchanged. This increase in the SpiB+ population may reflect that Sca1+ progenitor cells can give rise to both bronchiolar and alveolar cells to repair the damaged epithelium[53]. All populations had returned to naive levels by day 30, apart from a small increase in the number of SpiB+ club cells.

We also examined the expression of MHCII by the epithelial cells at the two infection timepoints. Ciliated cells were the most affected by the infection, with both SpiB-negative and positive populations more likely to express MHCII at day 10 and, while reduced, levels were still significantly increased at 30 post-infection compared to cells from naive mice (Fig. 4D). The amount of MHCII expressed by MHCII+ SpiB+ epithelial cells subsets was measured via MHCII geometric MFI (Fig. 4E). SpiB-negative and positive cells that were MHCII+ expressed broadly similar levels of MHCII. SpiB-negative and positive Sca1+ progenitor cells expressed higher levels of MHC II at day 10 but not day 30 post-infection, compared to cells from naive mice. In contrast, SpiB+ club cells expressed increased levels of MHCII at day 10, and these levels were still increased at day 30 post-infection. We also found small but significant increases of MHCII on CD24+ progenitor and ATII populations at day 30 post-infection compared to levels in cells from naive mice.

Together, these data show that SpiB+ epithelial cells form a large proportion of the upper and distal airway epithelium and that while SpiB+ cells are more likely to express MHCII, SpiB is not required for MHCII expression. IAV infection had at most a minor lasting impact on the numbers of different epithelial cell subtypes, but did lead to a sustained increase in the proportion of SpiB+ ciliated cells that expressed MHCII and increased levels of MHCII on several different epithelial cell subsets.

## Influenza virus infection leads to the sustained presence of populations of epithelial cells expressing high levels of MHCI and MHCII

We also examined whether the expression of the MHCI was increased by infection. As all nucleated cells express MHCI, we quantified the amount of the MHCI molecule $K^b$, on the epithelial cells. SpiB positive cells have a higher forward scatter than SpiB negative cells (Fig. 7C). As the anti-$K^b$ was labeled with Alexa 488, we found that SpiB positive cells had a higher autofluorescence than SpiB negative cells on the FTIC channel (SFig. 8A). Therefore, the data are normalized by dividing the MFI on each population by its FSC MFI.

In naive mice, there were significant but very small differences in MHCI expression between SpiB-negative and SpiB-positive populations (SFig. 8B). The most notable change in MHCI expression was at day 10 following infection, particularly in ciliated, club, and Sca1 progenitor cells. By day 30, MHCI expression was reduced, and while we found significant differences in comparison with cells from naive mice, it was difficult to interpret whether these small differences were biologically meaningful.

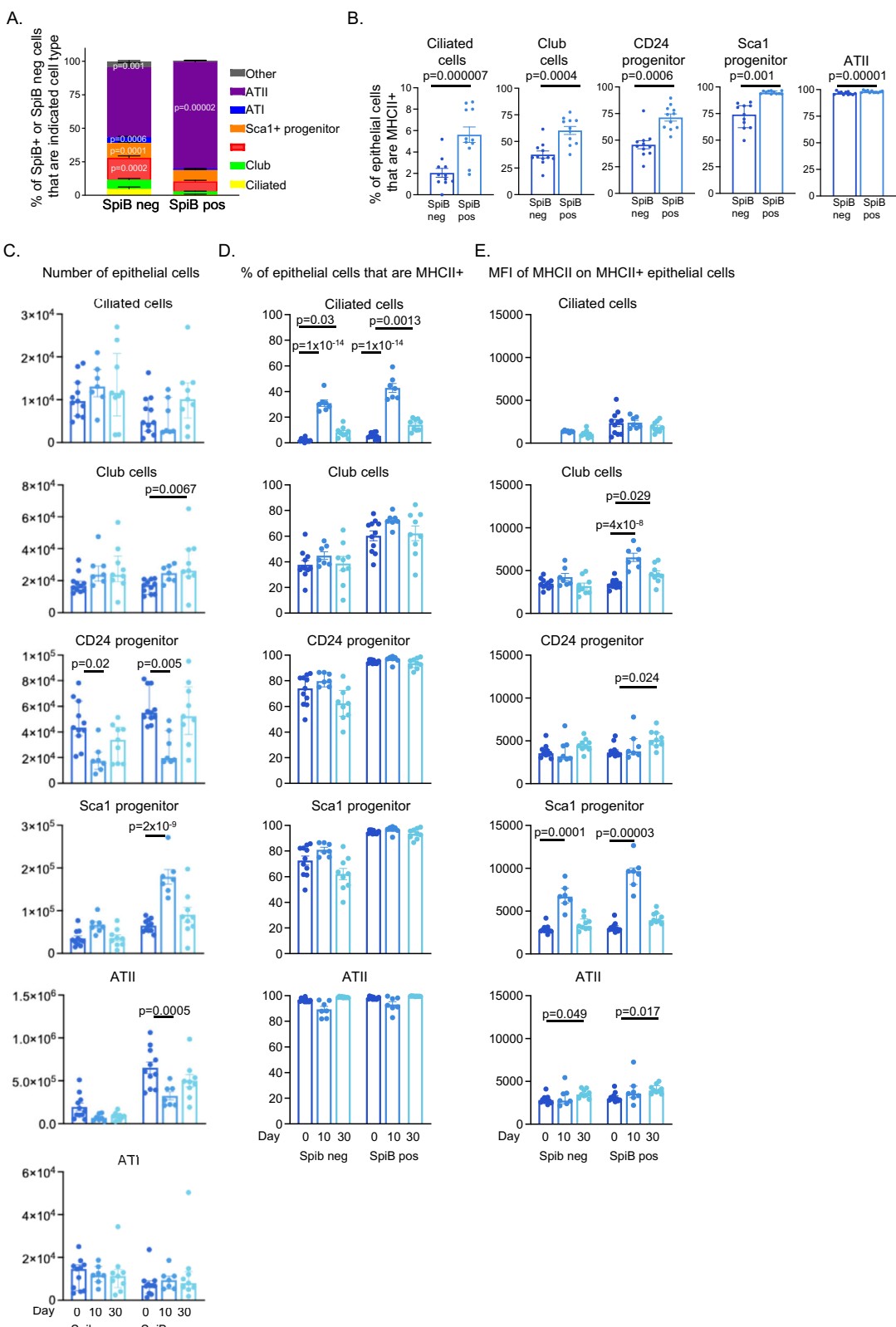

Instead, we gated on epithelial cells that expressed high levels of both MHCI and MHCII, as these cells would be able to communicate with both CD8 and CD4 T cells. Less than 1% of SpiB-negative or positive epithelial cells of any type expressed high levels of the MHC molecules in naive mice, making comparisons between these populations biologically difficult to interpret. IAV infection led to substantial increases in this population at day 10 in all epithelial cell populations examined apart from ATI cells (Figs. 5A, B, S8C). This change was most notable in ciliated, club, and Sca1+ progenitor cells, with over 50% of their SpiB+ populations expressing high levels of MHCI and MHCII. The percentages of MHCI-II high cells declined by day 30, but >1–13% of the SpiB+ ciliated, club and Sca1+ progenitor cells continued to express high levels of MHCI and MHCII.

**Fig. 4 | SpiB+ epithelial cells express more MHCII than SpiB-negative cells.** SpiB-mCherry reporter mice were infected with influenza A virus (IAV) (100-200PFU) and lungs taken at day 0, 10, or 30 post infection. EpCAM1+ epithelial cell populations were identified by gating on lineage (CD45, CD31, and CD140a) negative, SpiB negative or SpiB+ populations. **A** Frequency of SpiB positive or negative lung epithelial cells by cell subset in naive mice; the *p* values indicate that the population is present at a significantly greater percentage within the SpiB negative or positive subsets the color code is indicated in the figure. **B** Frequency of MHCII+ cells by each epithelial cell subset in naive mice. **C** Numbers of SpiB negative or positive populations at the different timepoints. **D** Percentage and **E** MFI of each population expressing MHCII at day 0, 10 and 30 post infection. In all experiments, data are combined with $N = 11$ naive, $N = 7$ primary, and $N = 10$ memory from two independent experiments. **A** Data show mean with SEM and differences tested by a one-way ANOVA and Šidák's multiple comparison test with *p* values located in the group in which the cell type is present at a higher frequency. **B–E** each symbol represents a mouse and the bars show means with SEM error bars for normally distributed data and median with interquartile range for non-normally distributed data. **B** All data are normally distributed apart from Sca1+ progenitor cells. **C** All data are not normally distributed apart from Sca1 progenitor and Alveolar type (AT)II cells. **D** All data are normally distributed apart from Sca1 progenitor cells. **E** All data are not normally distributed apart from ciliated and club cells. **B** Normally distributed data are tested via paired *t* tests and non-normal by a paired Wilcoxon test. **C**, **D** normally distributed data are tested via a one-way ANOVA with a Šidák's multiple comparison test and non-normal by a one-way Kruskal–Wallis test followed by a Dunn's multiple comparisons test. Source data for all panels are provided as a Source Data file.

Together, these data indicate that several different epithelial cell populations alter their expression of MHCI and MHCII following infection and that some differentiated and progenitor populations continue to maintain altered expression for at least 30 days following infection.

### Epithelial cells from IAV re-challenged mice have less virus than those from primary infected animals

We next investigated the consequences of a second IAV infection on lung epithelial cells and fibroblasts. To reduce the impact of neutralizing antibody responses, we infected mice with two different strains of IAV, WSN (H1N1) in the first infection and X31 (H3N2) in the re-challenge. We compared the response in the re-challenged mice with a primary X31 infection in age-matched naive animals.

To identify infected epithelial cells and fibroblasts by flow cytometry, we used a fluorescently labeled anti-IAV nucleoprotein (NP) antibody. Mice infected 30 days earlier with WSN had reduced percentages and numbers of IAV-NP+ epithelial cells two days after infection with IAV-X31 compared to infected naive mice (Fig. 6A).

To examine this difference in more detail, we gated on the epithelial cell subsets and examined the percentages of these cells that were IAV-NP+ (Fig. 6B). In primary infected animals, ciliated and Sca1+ progenitor cells were the most likely to be infected, followed by club cells. Very few CD24+ progenitor and ATII cells were infected, and too few ATI cells were detected to examine. Prior infection led to reduced IAV-NP in ciliated, club, and a small decline in ATII cells, but no differences in the progenitor cells.

Few fibroblasts (CD140a + ) were IAV-NP+ after IAV infection and there were too few cells detected to examine this further (SFig. 9A). The number of interferon-responsive[3] fibroblasts were reduced in re-infected mice compared to primary infection (Fig. 6C, D and gating in SFig. 9B). These data highlight that while fibroblasts are much less likely to be infected than epithelial cells, they are susceptible to local microenvironmental changes.

To determine if the reduction in IAV-NP levels by lung structural cells could be explained by enhanced/rapid phagocytosis of the virus by immune cells, IAV-NP levels were measured in lung myeloid and B cells. We found no evidence for this; all cells contained equivalent levels of NP apart from conventional Dendritic Cells (DC) 1, which contained less NP in the re-infected compared to the primary infected animals (SFig. 10A, B).

### Re-challenge with IAV leads to rapid upregulation of genes involved in communication with T cells by lung epithelial cells

To investigate the consequences of re-infection in more depth, we compared gene expression in FACS-sorted EpCAM1+ cells 2 days after a primary or re-challenge infection. EpCAM1+ cells from re-infected mice had increased expression of molecules in class I and II antigen presentation pathways (*B2m, H2-K1, H2-DMB2, Ciita*) and virus detection (*Nlrc5*) (Fig. 7A, B and Supplementary Data 4). Detection of IAV-NP+ cells requires fixation, making it difficult to investigate gene expression in FACS-sorted cells. Therefore, we investigated whether gene expression is different in infected cells in primary infected and re-infected animals by combining RNAscope with GeoMx spatial transcriptomics. First, we identified IAV-infected airways in primary and re-infected animals using a probe for IAV-X31 (Fig. 7C). Only three of the five re-infected mice that we analyzed contained virus+ areas, and we found a reduction in areas of lung containing virus in re-infected compared to primary infected animals (SFig. 11A). We then examined gene expression by GeoMx in virus+ airway areas, and virus-negative areas either in the same airway as the infected cells, or from adjacent airways (Supplementary Data 5 and 6). We also included airways from sections of the lungs that did not contain any virus taken from primary and re-infected animals.

As expected, in primary infected mice, a comparison of gene expression in virus+ areas versus airways from non-infected sections demonstrated an upregulation of anti-viral genes including *Cxcl10*, *Isg15* and *Irf7* and upregulation of pathways related to 'defense to viruses' and 'type 1 IFN' (SFig. 11B, Supplementary Data 7). An anti-viral response was also evident in virus-negative areas within infected airways and adjacent virus-negative airways. Similar to the primary infected mice, in re-infected animals, genes within 'type 1 IFN' and 'response to virus pathways' were upregulated in virus+ airways compared to airway cells from non-infected sections from the same mice (SFig. 11C, Supplementary Data 7).

To compare the responses between primary and re-infected animals, we performed two types of analysis. First, we counted the number of upregulated genes within the Gene Ontogeny (GO) terms 'Inflammation' and 'Type 1 IFN' identified in virus+, virus-negative areas within virus+ airways (close), and virus-negative adjacent airways (further) and calculated these as a percentage of the total DEG within each area (Fig. 7D, Supplementary Data 7).

All three areas had genes within both GO terms in primary infected animals, although further areas had the lowest number. In contrast, in re-infected animals, while genes within the GO term 'Inflammation' were upregulated in virus+ and 'close' airways, no genes in this term were found in 'further' areas and no genes in GO term 'Type 1 IFN' were found in 'close' or 'further' areas in re-infected animals. These data suggest that while IAV may induce inflammatory type 1 IFN responses in infected cells in primary and re-infected mice, the response is much more contained in re-infected animals.

Second, we compared gene expression between virus+ airways taken from primary or re-infected animals (Fig. 7E, Supplementary Data 7). We found increased expression of genes in the GO term 'Cellular Response to IFNγ' including *Ccl5* and *Cxcl9* in the airways of re-infected mice (Fig. 7F). These data suggest that IFNγ-producing immune cells act on infected cells that in turn upregulate molecules that enhance communication between immune and infected structural cells. Our previous study tracking in vivo IFNγ production using reporter mice, demonstrated an increase in IFNγ + NK cells and ILCs at day 3 following

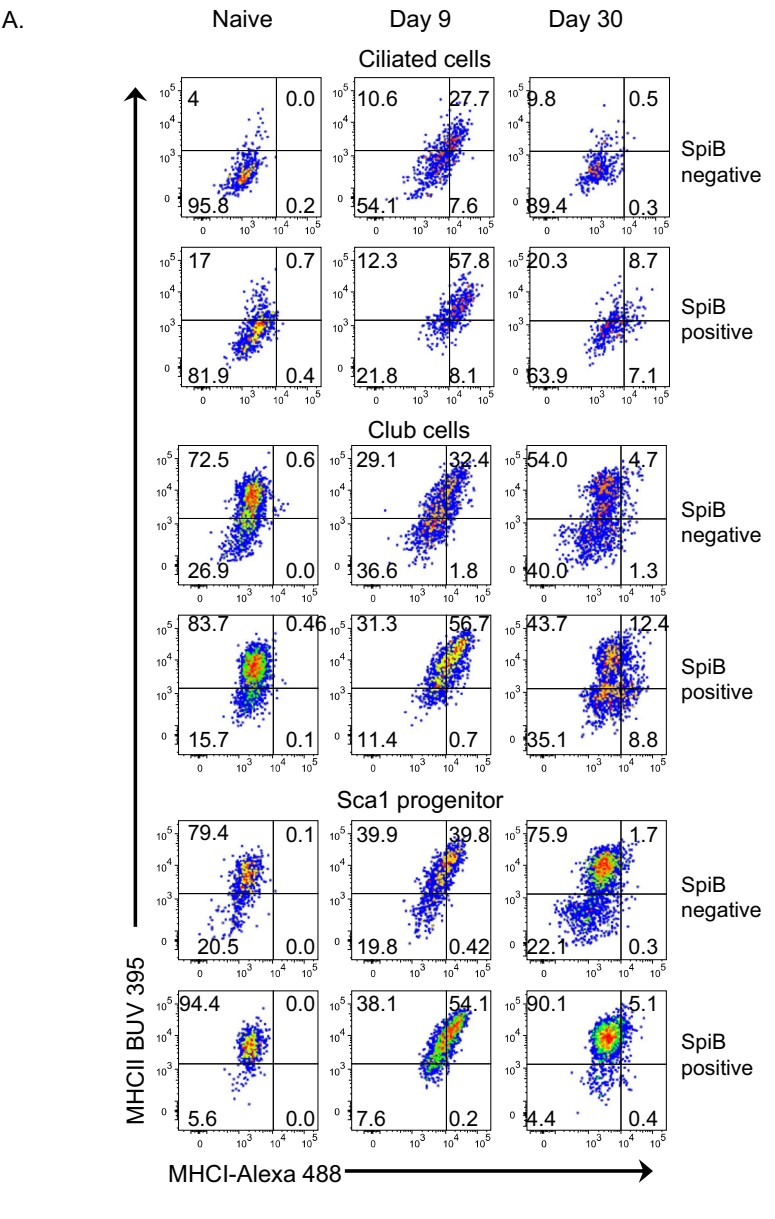

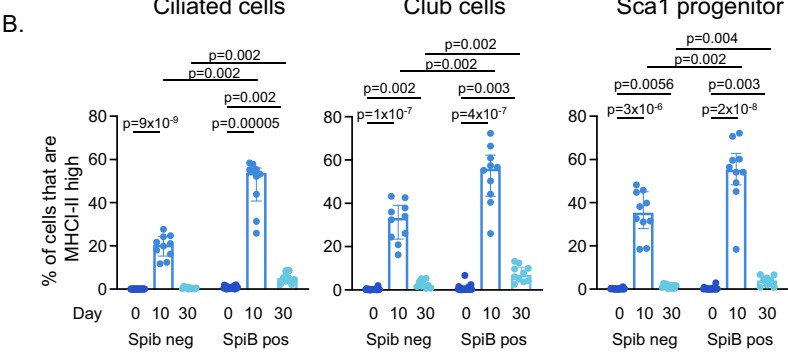

**Fig. 5 | Influenza virus infection leads to the sustained presence of populations of epithelial cells expressing high levels of MHCI and MHCII.** SpiB-mCherry reporter mice were infected with Influenza A Virus IAV-WSN (100-200PFU) and lungs taken at day 0, 10, or 30 post infection. EpCAM1+ epithelial cell populations were identified by gating on lineage (CD45, CD31, and CD140a) negative, SpiB negative or SpiB+ populations. **A** shows representative staining of MHCII and MHCI expression by ciliated, club and Sca1 progenitor cells at each timepoint. **B** Data are not normally distributed, each symbol represents a mouse and the bars show median with interquartile range. Differences between timepoints were tested by a one-way Kruskal–Wallis test followed by a Dunn's multiple comparisons test and separately between SpiB negative and positive populations by paired one-way Wilcoxon rank test. Data are from two independent experiments at each timepoint with a total of $n = 21$ naive; $n = 10$ day 10; and $n = 10$ day 30 mice. Source data for 5B are provided as a Source Data file.

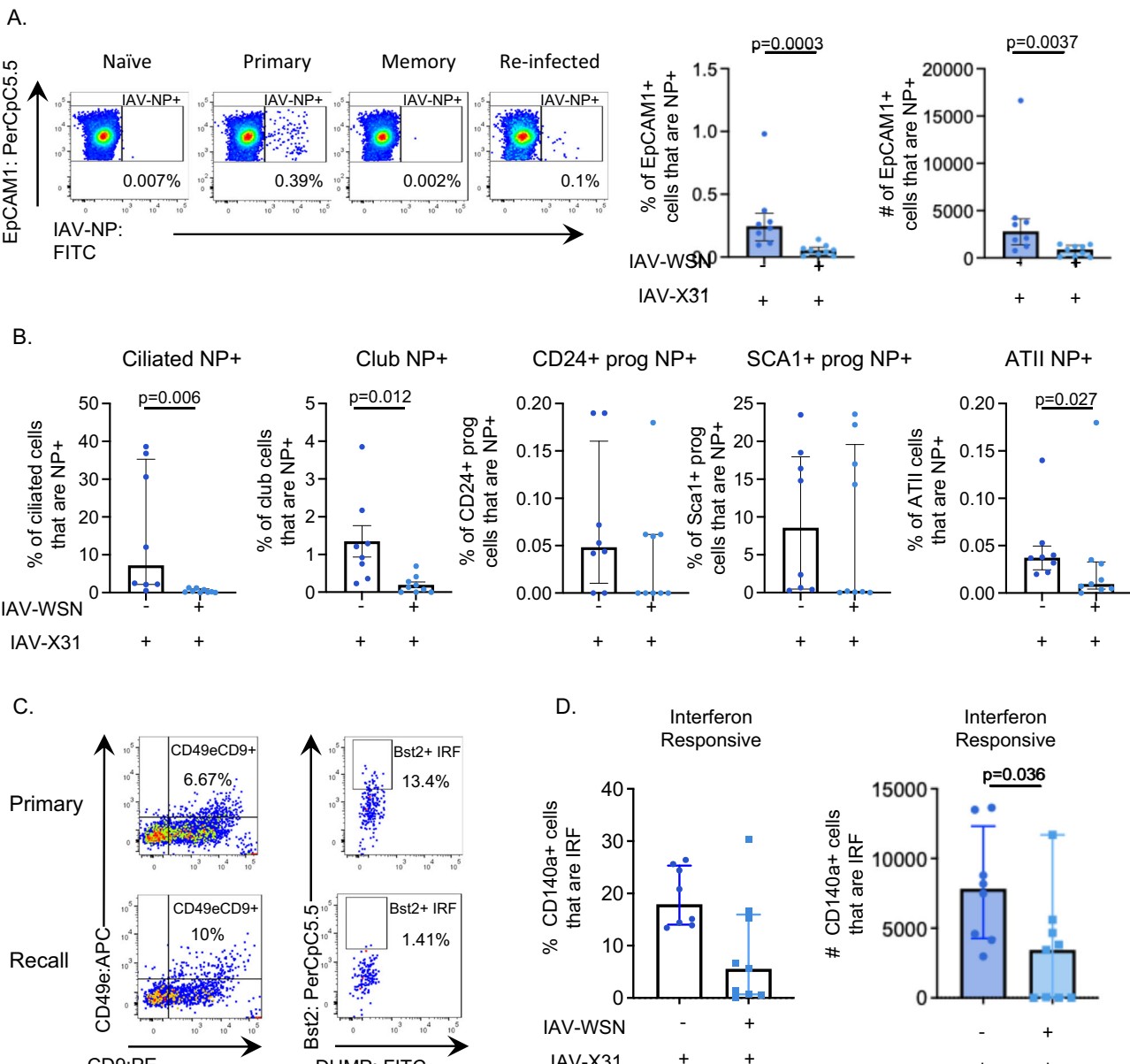

**Fig. 6 | Epithelial cells from IAV re-challenged mice have less virus than those from primary infected animals.** C57BL/6 mice that were either naive or infected 30 days earlier with influenza A virus (IAV)-WSN, were infected with IAV-X31, their lungs harvested 2 days later, and epithelial cells and fibroblasts examined by flow cytometry. **A** The percentages and number or EpCAM1+ cells positive for IAV-Nucleoprotein (NP) were examined by flow cytometry, gated as shown in SFig. 7C. **B** The percentages of each epithelial cell type that were IAV-NP+. **C** Representative FACS plots of IFN-responsive fibroblasts indicating positive populations, cells are gated on live, single, dump negative (CD45/CD31/EpCAM1) cells that are CD140a+ gated on CD49e+ CD9+ fibroblasts that are Bst2+ as shown in SFig. 9B. **D** the

percentages and numbers of IFN-responsive fibroblasts. Data are from two independent experiments combined. Primary: $n = 8$, re-infection: $n = 9$. In all graphs, each symbol represents a mouse, and the bars show means with SEM error bars for normally distributed data and median with interquartile range for non-normally distributed data. **A** Data are not normally distributed. **B** All data are not normally distributed apart from club cells. **D** The percentages and numbers of IFN-responsive fibroblasts are not normally distributed. In all graphs, normally distributed data tested via one-way $t$ test, and non-normally distributed data tested via one-way Mann–Whitney test. Source data for **A**, **B**, **D** are provided as a Source Data file.

re-infection[54]. In terms of number, conventional CD4 and CD8 T cells made up the largest population of IFNγ+ cells within the lung in re-infected animals, aligning with findings that IFNγ-producing CD4 and CD8 T cells accelerate virus control in mouse re-infection studies[21,55,56] and are associated with protection in humans[20,57–59].

### Infection-experienced structural cells more effectively control IAV than cells from naive mice

The potentially increased communication between infected airway cells and T cells indicated that, at this early time point following

infection, the enhanced protection provided by a prior IAV-infection will be dependent on T cells. To test this, we infected two cohorts of C57BL/6 mice with IAV-WSN and treated one cohort with IgG and the second with depleting CD4 and CD8 antibodies (SFig. 12A). Mice were treated 2 days prior and on the day of the infection with X31; control experiments demonstrated a loss of CD4 and CD8 T cells in lymphoid organs and the lung at day 2 post-re-infection (SFig. 12B). In data consistent with Epstein et al. [29], we found that mice treated with IgG and T cell depleting antibodies had less virus in their lungs compared to primary infected animals at day 2 post-infection (Fig. 8A). Moreover,

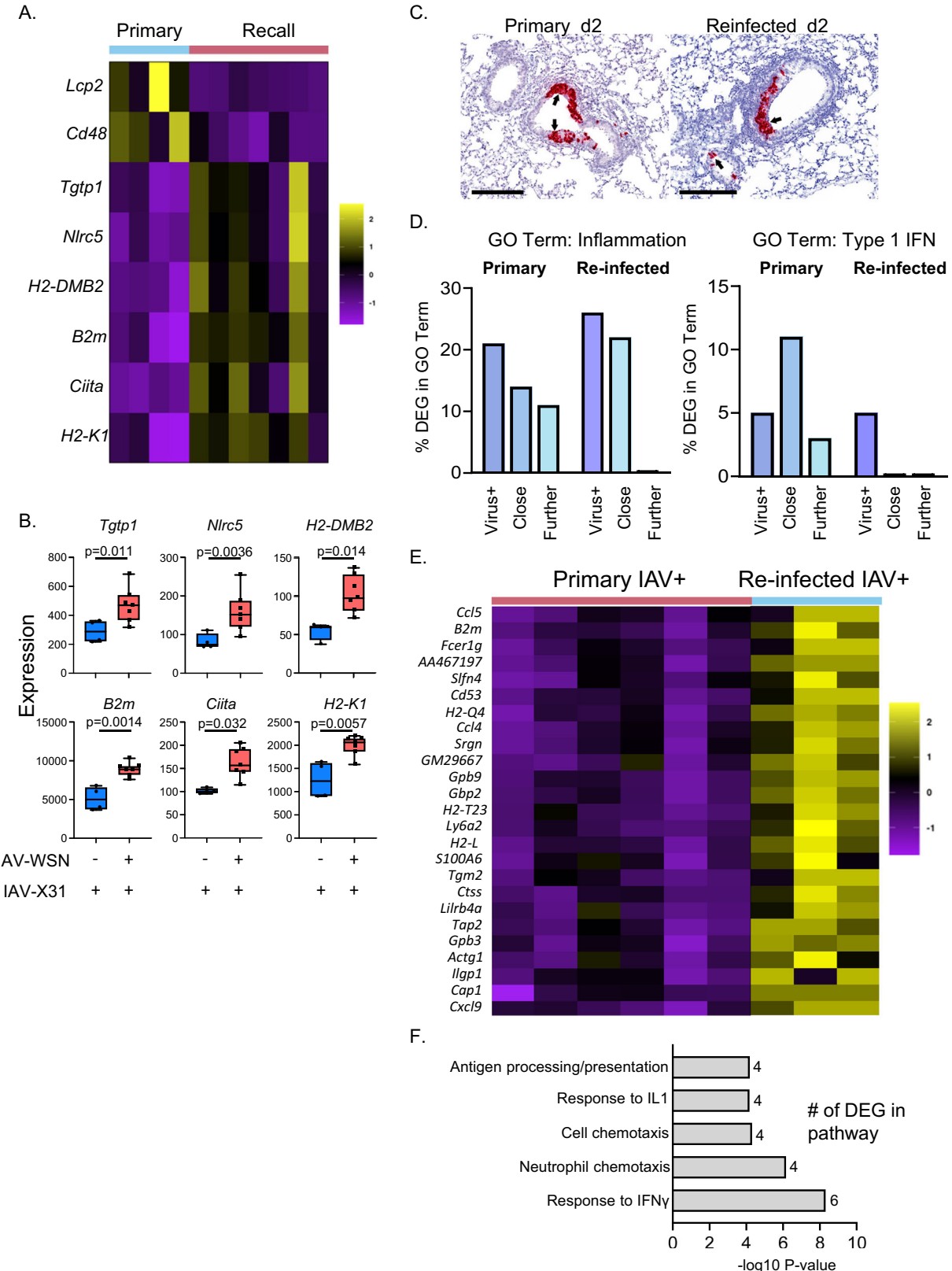

there was no difference in levels of virus between the two re-infected groups. These data suggest that T cells are not required for enhanced protection at day 2 post-infection and that, potentially, enhanced viral control responses are from the infected cells themselves.

We chose to use a reductionist in vitro model to investigate this finding further, reducing the complexity of a rapidly changing response in vivo (Fig. 8B). First, we asked whether the prior infection altered the ability of the structural cells to present antigen to CD4 and CD8 T cells. To achieve this, we isolated CD45-negative cells from naive and day 30 IAV-WSN-infected animals and rested these overnight before infecting them with IAV-X31 in vitro. The following day, T cells were isolated from the spleens of mice infected with IAV (X31; day 9/10 post-infection) to generate a pool of IAV-specific CD4 and CD8 T cells. These cells were co-cultured with the infected CD45 negative cells, and

**Fig. 7 | Spatial analysis of gene expression in primary and re-infected animals show altered inflammatory and type I IFN response and increased IFNγ response in re-infected animals.** C57BL/6 mice that were either naive or infected 30 days earlier with Influenza A virus (IAV-WSN), were infected with IAV-X31, their lungs harvested 2 days later, their epithelial cells sorted by FACS and their gene expression analyzed by Nanostring nCounter or lungs fixed and embedded in paraffin. **A**, **B** Heatmap and example boxplots showing Differentially Expressed Genes (DEG) from primary or re-infected animals in FACS sorted epithelial cells. DEG with a twofold change and an FDR *q* value < 0.05 in lung epithelial cells were considered statistically significant to take into account multiple comparisons. Primary group (*n* = 4) and re-challenge group (*n* = 7), data are from two independent experiments combined and each dot represents a mouse, the bounds of the boxes are the 25–75th percentiles, the lines in the boxes are at the median and the whiskers show the min and the max values. **C** RNAscope analysis for X31-IAV was performed on FFPE lung tissue; representative X31-IAV+ areas are shown in red and labeled with black arrows to highlight virus + bronchial epithelium. Bars, 200 micrometers. Virus-positive and negative airways from six primary and three re-infected mice were analyzed by GeoMX. **D** DEG between normal tissue and three areas: virus+ airway cells; virus negative cells within the same airway (Close); and adjacent virus negative airways (Further) were classified based on presence within MGI GO terms 'Inflammation' or 'Type 1 IFN'. **E** Heatmap showing DEG between X31-IAV positive airways in primary versus re-challenge groups, DEG with a twofold change and an FDR *q* value < 0.05 in lung epithelial cells were considered statistically significant. **F** Enriched pathways found in DEG identified between viral+ airways in day 2 primary or re-infected C57BL/6 mice analyzed by GeoMx. The *x* axis represents the −log10 of the *P* value given during the analysis; (FDR < 0.05 to take into account multiple comparisons) with the number of DEG listed from each pathway. Source data for **B**, **D**, and **F** data are provided as a Source Data file.

expression of T-cell activation markers, CD25 and CD69, was analyzed by flow cytometry after a further 24 hours.

CD4 and CD8 T cells co-cultured with in vitro IAV-infected CD45-negative cells from either naive mice or previously infected mice expressed higher levels of CD25 and CD69 than T cells co-cultured with control CD45-negative cells that did not receive IAV in vitro. This demonstrates that the structural cells could process and present IAV antigens to CD4 and CD8 T cells (Fig. 8C and S13A, B).

To confirm that this T cell activation was antigen dependent, we infected CD45-negative cells with IAV that express ovalbumen (OVA) peptides, either the SIINFEKL epitope (OVAI) recognized by OTI cells or OVA323-325 (OVAII) recognized by OTII cells. The infected cells were co-cultured with naive TCR transgenic OTI and OTII T cells for three days. We only observed upregulation of CD25 and CD69 on CD4 OTII T cells when the cognate antigen was present (SFig. 14A–C). Cognate antigen also drove substantial upregulation of CD25 and CD69 on OTI CD8 T cells. Additionally, we found a very small but significant increase in CD69 expression in OTI CD8 T cells cultured in the presence of IAV-OVAII, which was potentially driven by cytokines produced by activated CD4 T cells within the same well.

We expected that CD45-negative cells from previously infected animals might present antigens more effectively to T cells, given their increased expression of molecules involved in these processes. Expression of CD25 and CD69 were equivalent between splenic CD4 T cells from infected mice or naive OTII T cells cultured with IAV-infected structural cells taken from naive or IAV-infected mice (Fig. 8C). There was a slight but significantly lower expression of CD25 and CD69 on splenic CD8 T cells cultured with IAV-X31-infected structural cells from previously infected animals compared to X31-infected structural cells from naive mice. Expression of CD25 and CD69 was not significantly different but tended towards reduced expression in OTI cells co-cultured with structural cells from memory compared with naive mice (SFig. 14C).

Potentially, prior infection enabled the CD45-negative cells to control the virus more effectively, and thus these cells might contain lower levels of antigen. To test this, we harvested a separate plate of structural cells 24 hours after in vitro infection and examined the levels of IAV-NP in epithelial cells, fibroblasts and BECs by flow cytometry. Few BECs were infected with IAV; in contrast, clear populations of fibroblasts and epithelial cells were IAV-NP+ (Fig. 8D). More NP+ cells were found in infected cells compared to the non-infected controls. The exception was BECs from previously infected mice in which there was no significant difference between the no infection and infection control, suggesting these cells may be controlling the virus more effectively than cells from naive animals. However, there was also no significant difference between the two infected groups. In contrast, there were twice as many NP+ epithelial cells in cultures from naive mice compared to cells from previously infected animals. In the fibroblasts, there was no difference in the percentages of NP+ cells

between the infected groups. These data suggest a prior IAV infection enhances the ability of epithelial cells to control a subsequent challenge and demonstrate a cell-intrinsic effect, as fibroblasts within the same cultures were infected at the same rate, regardless of the cell source.

To ask whether this reduced infection led to a lower inflammatory response, we examined culture supernatants for the presence of cytokines and chemokines 24 h after the in vitro infection using a multiplex array. The supernatants from CD45 negative cells isolated from previously infected mice, contained higher levels of IL-1α, IL-12 p40 and p70, IL-13, IL-15, and CXCL9 compared to supernatants from cultures with cells from naive mice (Fig. 8E). Type I and III IFN were not detectable by ELISA (below the limit of detection; 31.3 pg/ml). In conclusion, these data suggest that the rapid control of the virus by the structural cells is accompanied by an increased inflammatory cytokine response, perhaps to warn and attract immune cells to control any virus not contained by the epithelial cells themselves.

## Discussion

Our functional and geographical analysis of the post-IAV lung demonstrates substantial and sustained changes to the three main structural lung cell types, epithelial cells, fibroblasts, and BECs. All three cell types maintain upregulated expression of genes involved in antigen processing and presentation, suggesting that they may have an enhanced ability to communicate with T cells. While interactions between T cells and lung structural cells may be important[46,60], we found that lung epithelial cells from mice previously infected with IAV can themselves control IAV more efficiently than cells from naive mice. These data demonstrate enhanced protective responses by epithelial cells, indicating these cells display protective immune memory.

Evidence for innate or trained immunity in multiple cell types has been growing over the last ten years[8,24,61]. Skin epithelial cells previously exposed to inflammatory stimuli respond more rapidly through an inflammasome-dependent mechanism[62], and lung epithelial cells that survive an IAV infection have altered gene expression and inflammatory responses following re-infection[25–27]. Fibroblasts are known to play important roles in lung anti-viral responses and in other organs, including synovial fibroblasts in joints and gingival fibroblasts, which can display evidence of memory to previous insults[61,63–66]. Most of the evidence for sustained changes in BECs comes from the field of cardiovascular disease, where chronic inflammation may sustain altered responses[61,67–69]. Our data extended the field by examining all three cell types within one organ, demonstrating a shared sustained response in all cell types, suggestive of coordinated tissue memory.

Epithelial cells, most notably AT-II cells, can express MHCII[46,60,70]. Antigen presentation by lung epithelial cells is required for normal responses to *Streptococcus pneumoniae* and Sendai virus infections in mice[46,60]. We found that IAV infection led to upregulation of MHCII either by the proportion of cells that were MHCII+ or increased

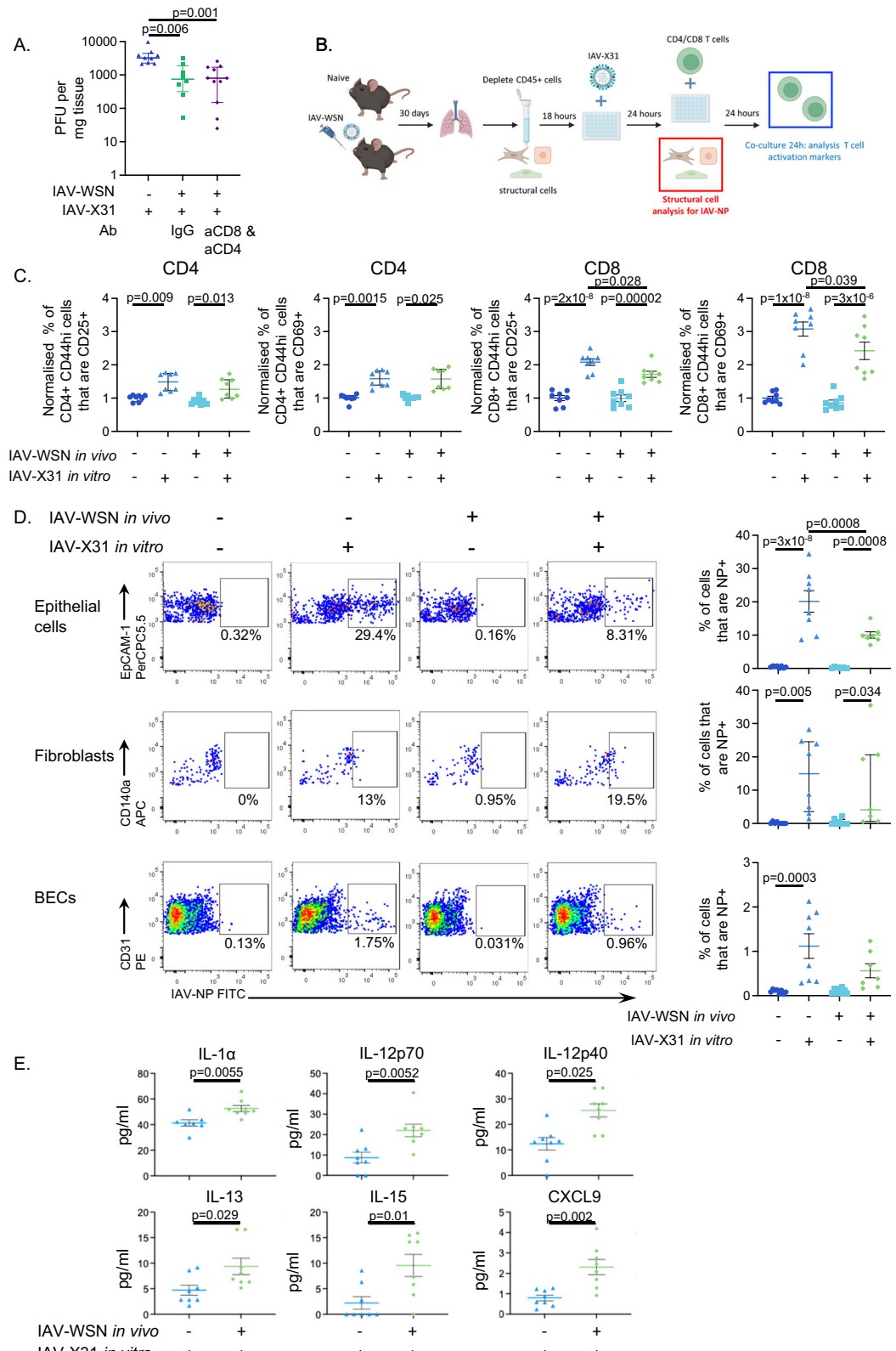

expression on upper (ciliated, club), progenitor (CD24+, Sca1+), and lower airway (ATII) epithelial cells at day 10 post-infection. Some of these changes, including more ciliated cells expressing MHCII, persisted at day 30 post-infection. As ciliated cells can live for months[71], our findings may indicate these MHCII+ ciliated cells persist from the infection, while other changes may be sustained by the progenitor epithelial cells.

Indeed, Fiege et al.[27] demonstrated that ciliated epithelial cells can survive IAV infection. Interestingly, other mouse models of epithelial lung injury[72] also reveal upregulation of MHCII by ciliated cells, and a recent human challenge study found a small subset of ciliated cells hyper-infected with SARS-CoV2 that were MHCII+ but that expressed anti-inflammatory molecules[73]. This study identified HLA-DQA2 (associated with MHCII antigen presentation) as a predictor of infection

**Fig. 8 | Infection-experienced lung structural cells display enhanced viral control and can reactivate T cells. A** C57BL/6 mice were infected with influenza A virus (IAV)-WSN on day 0 and treated with 400µg isotype control or 200µg each of anti-CD4 (GK1.5) and anti-CD8 (2.43) on days 28 and 30 when mice were (re)-infected with IAV-X31. Lung IAV titers were examined 2 days post-IAV infection, and data were combined from two experiments: primary (*n*=9) mice, recall IgG (*n*=9), and recall aCD4/aCD8 (*n*=11). **B** Experimental design of IAV infection, in vitro re-challenge and co-culture, schematic created in BioRender; Worrell, J. (2025) https://biorender.com/hwavd50. Lung CD45-negative cells were isolated from naive or IAV-WSN-infected mice 30 days earlier. After 24 hours, the cells were infected with IAV-X31 and the cells were examined by flow cytometry after a further 24 hours or co-cultured with T cells from the spleens of day 9 IAV-X31-infected mice infected. **C** Percentage of CD44hi CD4/CD8 T cells that were CD25+ or CD69+ , normalized to the mean of the no infection control within each of the two experiments, *n* = 4 mice per group in each experiment. **D** Representative FACS plots and data for IAV-Nucleoprotein (NP) for each cell type, graphed data are the average of two technical replicates per mouse. Cells were gated on live, single, CD45-negative EpCAM1+

(epithelial cells), CD140+ (fibroblasts), and CD31+ (blood endothelial cells). Numbers in plots indicate the percentages of cells that are IAV-NP + . **E** Supernatants from the infected cultures were tested for multiple immune mediators by Luminex 24 hours after in vitro infection. Data in **D**, **E** combined from the same two experiments: Primary: *n* = 8, Re-infection: and *n* = 7; in **D** one sample in the IAV-memory/IAV-31 group was lost due to a technical error during acquisition. In all graphs, each symbol represents a mouse, and the bars show means ± SEM for normally distributed data and median with interquartile range for non-normally distributed data. **A** Data are not normally distributed. CD4 data are not normally distributed, and CD8 data are normally distributed. **D** Epithelial cell and BEC data are normally distributed, and fibroblast data are not normally distributed. **E** All data are normally distributed. **A**–**D** non-normally distributed data tested with a one-way Kruskal-Wallis test followed by a Dunn's multiple comparisons test. **C**, **D** normally distributed data tested by a one-way ANOVA followed by Šidák's multiple comparison test. **E** Data tested by a one-way t-test. Source data for **A**, **C**–**E** data are provided as a Source Data file.

---

outcome and protection[73]. Sustained elevated MHCI and MHCII levels may reflect an imprint of infection, providing these epithelial cells with an advantage by more quickly alerting T cells to a new infection. However, as non-professional APCs, the cells could drive T cell tolerance rather than activation[74].

We bioinformatically identified the transcription factor, SpiB, as a potential upstream regulator of the altered MHCII expression and upregulation of gene expression in multiple molecules involved in antigen processing and presentation[33]. In addition to B cells, SpiB is expressed by plasmacytoid dendritic cells[49,50], Micro-fold cells[28], tuft cells, some fibroblasts in secondary lymphoid organs[75], and some thymic epithelial cells[76]. Our data suggest that it is also expressed by multiple populations of lung epithelial cells and that SpiB+ positive cells express higher levels of MHCII. Recently, two studies identified SpiB+ epithelial cells in distinct anatomical locations. Surve et al. performed re-analysis of publicly available single-cell RNA-sequencing data and detected rare M cells in the homeostatic mouse trachea[77]. While Barr et al. used single-nucleus RNAseq to detect lung M cells expressing *Tnfaip2, Sox8, Spib, Ccl9, Ccl20*, and *Tnfrsf11a*[78].

We showed that following IAV infection, *Spib*+ upper airway epithelial cells were likely to be close to persistent clusters of immune cells. These observations are consistent with the study by Barr et al., where *Spib*+ cells in the IAV-infected lung persisted from day 5 to 56 post infection[78]. However, our study provides additional insight by validating this at the protein level and by examining MHCI and MHCII expression in naive and infected animals. We and others have characterized these clusters, which predominantly contain B and T cells[17,34–38]. The close proximity of these immune cells suggests that they may release molecules to sustain the altered epithelial cells, and/or their proximity evidences the past location of viral replication.

Our data support that fibroblasts can contribute to the early anti-viral response through multiple mechanisms, including communication with T cells. Although fibroblasts are much less likely to become infected with IAV than epithelial cells[10], several studies have highlighted their ability to drive immunopathology or restrain lung inflammation by communicating with lung T cells[79–81]. Boyd et al. showed that damage-responsive fibroblasts can influence the migration of CD8 T cells that enter inflamed tissues and that this amplifies immunopathology[3]. Conversely, amphiregulin-producing regulatory T cells promote alveolar epithelial regeneration and repair by activating Col14+ adventitial fibroblasts, supporting progenitor cells after IAV-induced damage[79]. Conditional deletion of these Col14+ fibroblasts reduced survival rates following IAV infection. Additionally, Treg cells secrete IL-1Ra, which suppresses IL-1 receptor-mediated chemokine production by adventitial fibroblasts, mitigating excessive inflammation[80]; disruption of this

regulatory axis caused exacerbated inflammation and impaired viral control. Our investigation adds further complexity, suggesting that previous viral infections could alter the consequences of T cell-fibroblast interactions characterized by these studies of primary infections.

A classic characteristic of immune memory is the ability to respond rapidly to re-infection[82]. By measuring gene expression and developing an in vitro infection model, we demonstrated that lung epithelial cells display an enhanced response to re-infection, leading to more rapid control of IAV. It is possible that multiple cell types can contribute to the reduced virus titer at day 2 following infection, these could include altered alveolar macrophages[83,84] and/or T cells[85]. We focused on structural cells as they, in particular epithelial cells, are the first cells infected with the virus and thus the first to make an anti-viral response. The reduction of the virus only in the epithelial cells in the in vitro model indicates a cell-intrinsic protective response in these cells.

The enhanced anti-viral response measured in vitro was accompanied by increased production of cytokines. In primary infected mice, Hamele et al. [86] identified a population of IAV-infected ciliated cells that produced the majority of inflammatory cytokines early in the response. Surviving ciliated cells may not only therefore express more MHCII, but may also be responsible for the enhanced cytokine response we found in in vitro infection of CD45-negative cells from previously infected mice.

Krausgruber et al.[87] found that even in naive animals, structural cells are poised, ready to produce molecules more often associated with immune cells. Our data extend this finding, suggesting that exposure to pathogens boosts this homeostatic immune gene expression, making structural cells an even more important component of protective immunity. However, we did not investigate epigenetic modifications to the structural cells, this is a known hallmark of trained immunity. Therefore, it is not possible to determine if the changes we observed are due to ongoing inflammation, potentially caused by ongoing antigen presentation, or the epigenetic rewiring that underpins trained immunity.

Our study also provides a rationale for attempting to target structural protective immunity as part of vaccine design, for example, with adjuvants. This concept was investigated in a recent study by Denton et al., who found that targeting TLR4 using adjuvants improved the MAdCAM-1 + stromal cell response to immunization and facilitated longer-term protection post vaccination[88].

Single-cell RNAseq experiments examining primary IAV-infection have identified that ciliated and club cells can exhibit transcriptional signatures of interferon signaling independent of infection but dependent on the inflammatory microenvironment[86,89,90]. Similarly, our spatial transcriptomic analysis (GeoMx) of mouse lungs from

primary infected mice identified that genes associated with the type I IFN response and inflammation were found beyond virus+ regions. These responses were more restricted in lungs from re-infected mice in which the anti-viral response was contained close to the virus. This potentially limits inflammation-driven immunopathology and the release of cytokines that cause systemic symptoms associated with the infection.

Both CD4 and CD8 T cells protect mice and humans from IAV re-infection, especially when priming and re-challenge strains express different hemagglutinin and neuraminidase proteins[18,20,21,29,30,57,58]. Studies in mice that have demonstrated a requirement for CD4 and/or CD8 T cells in heterosubtypic infection have mainly examined viral titers at day 5–7 following re-infection[21,30,91–93]. Slütter et al.[18] and Guo et al.[94] demonstrated a role for CD8 T cells as early as day 3 post-re-infection. However, both our data and that from Epstein et al.[29] show that at the earlier time point of day 2, T cells are not required for a reduction in viral titers.

Virus is still present at this timepoint and protection from re-infection is enhanced by T cells[21,29,94–97]. We found increased expression of inflammatory chemokines and molecules involved in the processing and presentation of antigen following re-infection by lung structural cells. These changes may lead to an enhanced ability to attract and communicate with local T cells. In summary, therefore, our data suggest that lung epithelial cells may play both early T cell-independent and then later T cell-dependent roles in accelerating viral clearance following re-infection.

## Methods

### Animals

All mice were housed and, where noted, bred at the University of Glasgow under specific pathogen-free conditions with a 12 hour light/dark cycle with lights on/off at 07.00/19.00, at 20–24 °C and 45–65% humidity. All protocols were carried out in accordance with UK Home Office regulations (Project Licenses P2F28B003 and PP1902420) and approved by the University of Glasgow Animal Welfare Ethical Review Body.

SpiB-mCherry reporter mice on a C57BL/6 J background were generated at MRC Harwell, United Kingdom. The sequence for a 2 A peptide (T2A[47]) with a GSG linker and the fluorescent molecule, mCherry, was inserted into the SpiB locus: Ensembl ENSMUSG00000008193. The knock-in was inserted at the end of the coding sequence. SpiB reporter mice were viable, fertile and born at expected Mendelian ratios. Male and female animals used in all experiments were bred at the University of Glasgow. In all experiments, mice were 9–14 weeks old at the start of the experiment. No formal randomization was performed; the sample size was based on previous studies, and sex was considered as a blocking factor in the experimental design.

OTI female mice were bred at the Cancer Research UK Scotland Institute and provided by Edward Roberts; OTII female mice were bred at the University of Glasgow and provided by James Brewer. Both strains of mice are on a C57BL/6 background, and mice were used at 12 weeks of age.

Ten-week-old female C57BL/6NHsd mice were purchased from Envigo (United Kingdom) and used at between 11 and 14 weeks of age.

### IAV infections

SpiB or C57BL/6 mice were briefly anesthetized using inhaled isoflurane and infected with 100–200 plaque-forming units (PFU) of IAV-WSN strain, depending on their age, weight, and sex or 200PFU IAV-X31 strain in 20 μL of phosphate-buffered saline (PBS) i.n. IAV was prepared and titered in Madin-Darby Canine Kidney cells (MDCK). Infected mice were weighed, and any animals that lost more than 20% of their starting weight were humanely euthanized. Cages with mice that lost weight were given a soft diet until all mice returned to their starting weight or above.

### S FLU virus

S-FLU virus[39] was obtained from the laboratory of Alain Townsend at the university of Oxford. C57BL/6 were infected with 5 HAU (hemagglutinating units) of either the parental strain of the virus IAV-WT X31 or the S-FLU-X31.

### T cell depletion experiment

Two cohorts of C57BL/6 mice were infected with IAV-WSN, and one cohort was injected with 400 μg of rat IgG (LTF-2) and the second with 200 μg CD4 (GK1.5) and 200 μg CD8 (2.43) antibodies, all antibodies from BioXCell. Mice were treated intraperitoneally with anti-CD4/CD8 2 days prior to re-challenge with IAV-X31 and 2 days following infection. Mice were culled 2 days post-re-challenge. Control IgG and anti-CD4/CD8 treatments were given to mice within the same cages to prevent cage-specific effects acting as confounders.

### Plaque assay on mouse lung homogenates

Once euthanized, mouse lungs were extracted and frozen at −80 °C. Lungs were thawed, weighed and homogenized with 1 ml of DMEM. Homogenates were cleared by centrifugation and titrated by plaque assay. Confluent monolayers of MDCK cells were washed twice with 'empty DMEM', infected with serial dilutions of virus and incubated for 1 h at 37 °C to allow virus adsorption to the cells. After virus removal, cells were overlaid with MEM with oxoid agar and incubated for 2 days at 37 °C, 5% $CO_2$. After removing the overlay, cells were fixed with PBS:4% formaldehyde and stained with a 0.08% Crystal Violet solution in 6.25% methanol solution for at least 20 minutes. Staining solution was rinsed under tap water, plates were air-dried, plaques were then counted by a blinded researcher.

### Tissue preparation

Mice were euthanized by cervical dislocation (RNA-sequencing, FACS sorting and flow cytometry analysis) or alternatively with a rising concentration of carbon dioxide (lung histology and imaging). For flow cytometry analysis of structural cells, single-cell suspensions of lungs were prepared by enzymatic digestion with a final concentration of 1.6 mg/mL Dispase, 0.2 mg/mL collagenase P (Roche, UK) and 0.1 mg/mL DNase (Sigma, UK) for 40 minutes at 37 °C in a shaking incubator and tissues disrupted by passing through a 100μm filter. Red blood cells were lysed with lysis buffer (ThermoFisher).

For analysis of CD4 and CD8 T cells by flow cytometry prior to euthanasia by cervical dislocation, mice were injected intravenously with 1 μg AF488-conjugated anti-CD45 (clone, 30F11, ThermoFisher) and organs harvested after 3 minutes. Single-cell suspensions of lungs were prepared by digestion of snipped lung tissue with 1 mg/mL collagenase and 30 μg/mL DNase (Sigma) for 40 minutes at 37 °C in a shaking incubator, and tissues were disrupted by passing through a 100 μm filter. Spleens and lymph nodes were processed by mechanical disruption. Red blood cells were lysed from the spleen and lungs with lysis buffer (ThermoFisher). Cells were counted using a hemocytometer, with dead cells excluded using Trypan Blue.

### Flow cytometry staining

Cells were incubated for 10 mins with Fc block (homemade containing 24G2 supernatant and mouse serum) surface stained with anti-CD45 FITC (Biolegend, 30-F11) or anti-CD45-PE (BD, 30-F.11) or anti-CD45 eflour450 (ThermoFisher, 30-F.11), anti-CD31 PE (Biolegend, MEC13.3) or anti-CD31-FITC (BioLegend, MEC13.3), CD326/EpCAM1 PerCP-Cy5.5 (Biolegend, G8.8) or CD326/EpCAM1 BV711 (Biolegend, G8.8), anti-CD140a/Pdgfra APC (Biolegend, APA5) or anti-CD140a/Pdgfra BV605 (Biolegend, APA5), Gp38/podoplanin PeCy7 (Biolegend, 8.8.1), anti-Sca1 APC-Cy7 (BioLegend, W18174A), anti-CD24 BV421 (Biolegend, M1/69), anti-MHCII BUV395 (BD: 2G9), anti-Kb-Alexa 488 (BioLegend, AF6-88.5), anti-Siglec F-APC (BioLegend, S17007L), anti-CD11b-PeCy7 (Biolegend, M1/70), anti-Ly6G-BV785 (BioLegend, 1A8), anti-Ly6C-

PerCP-Cy5.5 (ThermoFisher, HK1.4), anti-CD64-BV711 (BioLegend, X54-5/7.1), anti-CD11c-eFluor780 (ThermoFisher, N418), anti-B220-eFluor 450 (RA3-6B2, ThermoFisher), anti-CD103-Qdot 605 (BioLegend, 2E7), anti-Bst2-PerCp-C5 (BioLegend, 927) and anti-CD9-PE (BioLegend, MZ3). Cells were stained with a fixable viability dye eFluor 780 or eFluor 506 (both ThermoFisher) as per the manufacturer's recommendations. Cells were fixed with Cytofix/Cytoperm (BD Bioscience) when required for 20 min at 4 °C and stained in PermWash buffer with anti-IAV-NP FITC (Invitrogen, D67J) for 1 h at room temperature. Antibody dilutions are listed in the Reporting Summary.

For T cells, surface stains were anti-CD4 APC-Alexa647 (RM4-5, ThermoFisher), anti-CD8 BUV805 (BD 53-6.7), anti-CD44 BUV395 (BD, IM7), anti-CD25 BV711 (Biolegend, PC61), anti-CD69 PE (BD, H1.2F3), anti-B220 eFluor 450 (RA3-6B2), anti-MHCII eFluor 450 (M5114) and anti-F480 eFluor 450 (BM8), all ThermoFisher. Anti-B220, MHCII and F480 were used as a 'dump' gate. For the OTI/II experiment, additional antibodies used were Va2-PeCy7 (BD, B20.1), Vb5.1/5.2eFluor450 (ThermoFisher, MR9-4), CD25-APC (ThermoFisher, PC61.5) and CD8-BV785 (BioLegend, 53-6.7). Samples were acquired on a BD Fortessa and analyzed using FlowJo (version 10 BD Bioscience).

### Lung digestion and isolation of mouse lung structural cells by fluorescence-activated cell sorting
For FACS isolation of lung structural cells, single-cell suspensions were generated by digesting snipped lungs with a final concentration of 3.2 mg/mL Dispase-II, 0.4 mg/mL Collagenase P, and 0.2 mg/mL Dnase-1. Lung samples were incubated at 37 °C for 25 minutes. Red blood cells were lysed using RBC buffer (ThermoFisher). Cells were simultaneously stained with CD45 microbeads (Miltenyi Biotec) and antibodies: anti-CD45 FITC (Biolegend, clone:30-F11) anti-Ter119-FITC (BioLegend, TER-119), anti-CD31 PE (BioLegend, MEC13.3), CD326/EpCAM1 PerCP-Cy5.5 (Biolegend, G8.8), CD140a/Pdgfra APC (Biolegend, APA5). Hematopoietic cells were depleted using the LS MACS (Miltenyi Biotec) column as per manufacturer's instructions. CD45-depleted lung cells were stained with eFluor 780 viability stain (Thermofisher), and cells were sorted on BD FACS Aria IIU sorter. Cells were either sorted directly into RLT or first sorted into 50% FCS/RPMI, washed with PBS and then lysed in RLT buffer.

### RNA extraction for qPCR and nanostring analysis
RLT suspensions were centrifuged through Qiagen QIA-Shredders, and RNA was extracted from sorted cells using the RNeasy mini kit (Qiagen, UK) with the following changes to manufacturer's instructions: input to the RNeasy column was at a ratio of 100 μL sorted cells: 350 μL RLT: 200 μL 100% EtOH. Following the extraction, the samples were DNase-1 treated. For qPCR analysis, briefly, the first-strand cDNA synthesis was conducted using a Reverse Transcription System kit according to the instructions of the manufacturer (Promega).

### qPCR analysis
Gene expression was analyzed by qPCR (SYBR Green FastMix Quanta Bioscience) on a QuantStudio 7 flex and expression was calculated using standard curves and results normalized to *Gapdh* or *18 s* expression. qPCR standards were prepared using spleen or lung cells from IAV infected mice and purified by gel extraction (Quick Gel Extraction kit, Invitrogen) or PureLink PCR Purification kit (Invitrogen) and primer sequences are listed in Table 1.

### NanoString nCounter analysis of sorted epithelial cells
Briefly, RNA from sorted lung epithelial cells of either naive or IAV-infected mice (day 30) was analyzed using the nCounter Mouse Immunology Panel (catalog # 115000052 XT_PGX_MmV1_immunology). RNA was purified using an RNA Concentration and Clean-Up kit as per manufacturer's instructions (Zymo, Cambridge Biosciences, R1013). The integrity and quantity of total RNA were determined using

Agilent 2100 Bioanalyzer (Agilent Technologies); samples with a RIN value of ≥8 were included for downstream analysis. This panel includes 547 genes covering core pathways and processes of the immune response, and 14 internal reference genes, a custom code set of 29 genes was used in conjunction with this panel (Supplementary Data 2). Each sample consisted of RNA from sorted epithelial cells from one individual mouse. Assays were performed using 50 ng input RNA and quantified on the nCounter FLEX system, sample preparation station, and digital analyzer (NanoString Technologies) according to the manufacturer's instructions. Briefly, reporter and capture probes were hybridized to target analytes for 20 h at 65 °C as per ref. 98. After hybridization, samples were washed to remove excess probes. Purified target-probe complexes were aligned and immobilized onto the nCounter cartridge, and the transcripts were counted via detection of the fluorescent barcodes within the reporter probe. Raw gene expression data were analyzed using NanoString nSolver analysis software, version 4.0. Background subtraction was performed using the included negative controls included using the default threshold settings. Data normalization was performed using 6 house-keeping genes (*Gapdh, GusB, Hprt, Oaz1, Polr2, and Sdha*); these genes had been selected based on stability of expression levels from previous transcriptomic experiments. Ratios of transcript count data were generated for primary infected epithelial cells (day 2) versus IAV-re-challenged epithelial cells (day 30 + 2). Genes with an FDR value < 0.05 were considered statistically significant.

### RNA sequencing
Samples were pooled from 6 to 8 mice in each group. Cells were sorted into 50% FCS/RMPI, centrifuged and washed 1× with PBS, and 350 l μL of RLT was added to the cell pellets. Samples were centrifuged through a Qiashredder and stored at −80 °C. RNA was extracted using mini or micro-RNA kits as per manufacturer's instruction. RNA was purified from total RNA (100 ng) using poly-T oligoattached magnetic beads (Life Technologies). Sequencing libraries were generated using NEB Next Ultra Directional RNA Library Prep Kit for Illumina (NEB) in accordance with the manufacturer's recommendations. Products were purified and quantified using the Agilent high-sensitivity DNA assay on the Agilent Bioanalyzer 2100 system. After the clustering of the index-coded samples, each library preparation was sequenced on an Illumina NextSeq™ 500 platform with 2 × 75bp paired-end sequencing. An average sequencing depth of 20 M reads per sample was achieved. Reads were aligned to the mouse genome, mm10, with STAR version 2.4.2a (https://doi.org/10.1093/bioinformatics/bts635) in two-pass mode, and on the second pass reads mapping to genes were quantified. Both the genomic DNA sequence (Mus_musculus. GRCm38 primary assembly) and gene annotations (GTF file GRCm38.86) were downloaded from Ensembl (Zerbino et al. https://doi.org/10.1093/nar/gkx1098). Quality control of the samples was carried out with FASTQC (bioinformatics.babraham.ac.uk/projects/fastqc) and FastP used for trimming, Hisat2 for aligning and FeatureCounts to count the reads aligning to the gene regions. Differential expression analysis was performed with DESeq2, and a *p* value of <0.05 was considered statistically significant. Further sample quality control, visualization and exploration were performed in R via SearchLight2[99].

### Over-representation analysis and network analysis
For ORA, significantly different genes were compared to a database (STRING11.5) of pre-defined lists of genes. A hyper-geometric test was then used to determine whether each gene set is enriched or not for the significantly differential genes. We examined the 10 most enriched gene sets in the significantly upregulated genes. The false discovery rate (FDR) was calculated to correct the *p* value. The significant ORA terms were defined as an FDR < 0.05. For Network Analysis, enrichment analysis was performed using Hyper-geometric Gene Set Enrichment on the gene set databases STRING11.5. For each network,

**Table 1 | Primer sequences**

| Standard primers | Forward | Reverse |
|---|---|---|
| Gapdh | AAGGGCATCTTGGGCTACAC | TAAAGTTCGCTGCACCCACA |
| SpiB | TTGCTCTGGAGGCTGCA | GGTGAGTTTGCGTTTGACCT |
| Psmb9 | GGGACAACCATCATGGCAGT | TCTGGCCATGAACCGAGATG |
| H2-Ab1 | CAGGTGTGAGTCCTGGTGAC | AGTGTTGTGGTGGTTGAGGG |
| Tap1 | CGTCCAGATGCCTTCGCTAT | TCCTCAGTCACCCGAGATGT |
| Cd74 | GCCACCACTGCTTACTTCCT | TCTTCCAGTTCACGCCATCC |
| Ifi47 | CGCTTCAGCCTCAATGATGC | AAGGCAGAAGATTCAGGGCC |
| 18 s | CGTAGTTCCGACCATAAACGA | ACATCTAAGGGCATCACAGACC |
| qPCR primers | Forward | Reverse |
| Gapdh | AACTCCCACTCTTCCACCTTC | AAGGAGTAA-GAAACCCTGGACC |
| SpiB | GATGGCCCACACTTAAGCTG | AGCAGGATCGAAGGCTTCAT |
| Psmb9 | ATAGTAGCTGGCTGGGACCA | AGAACCGCCGATGGTAAAGG |
| H2-Ab1 | TGCTACTTCACCAACGGGAC | ACGTACTCCTCCCGGTTGTA |
| Tap1 | TCTCTCTTGCCTTGGGGAAAT-G | AGCCATATGTTGCGGGTGAA |
| Cd74 | GCCACCACTGCTTACTTCCT | CAGGCCCAAGGAGCATGTTA |
| Ifi47 | TGGGCACAAGCGTCATATGT | AAGGCAGAAGATTCAGGGCC |
| 18 s | GACTCAACACGGGAAACCTC | TAACCAGACAAATCGCTCCAC |

nodes represent gene sets and edges represent two gene sets with a Szymkiewicz-Simpson coefficient of at least 0.5.

## RNAscope in situ hybridization

Naive wildtype and IAV-infected mice were euthanized by cervical dislocation, and the lungs were removed. Briefly, lung tissues were fixed with a 10% neutral buffered formalin solution (Sigma-Aldrich, UK) for 24–36 hours and subsequently embedded in paraffin. Lung tissue was cut with a microtome (Leica) at 6 μm per section and mounted on Superfrost slides (Fisher Scientific). Slides were stored at R/T until use, then baked at 60 °C for one hour prior to commencing the RNA scope protocol. Pre-treatment of tissue sections and RNAscope ISH was carried out according to manufacturer protocols (Advanced Cell Diagnostics, Inc.) with protease for 30 mins, followed by heat-induced target retrieval, using the RNAscope Red 2.5 kit (ACD-Biotechne; Cat# 322350) with Mm-SpiB (408789), DapB (Cat# 310043) and Mm-Ubc (Cat# 310771). For X31-IAV RNAscope, probe 1226621-C1 (ACD) was used; slides were baked at 37 °C overnight and pre-treated, as above, combined with boiling for target retrieval.

## Spatial transcriptomics using nanostring GeoMx digital spatial profiler (DSP)

X31-IAV RNAscope was used to select virus-positive and negative areas from FFPE lungs of each mouse, and these cores were combined to generate a Tissue Microarray. Two serial 5μm thick slices were cut, one slide tested to confirm virus presence/absence by RNAscope and the second slide used for GeoMX DSP. The GeoMx DSP experiment consisted of slide preparation, with the nuclei stained with Syto13, tissue hybridization with UV-photocleavable probes (Mouse Whole Transcriptome Atlas panel of probes corresponding to 19,963 genes, Nanostring), slide scanning, region of interest (ROI) selection, probe collection, library preparation, sequencing, data processing and analysis. Detailed slide preparation has been previously described[100].

## ROI and area of illumination selection and probe retrieval.

Slides were scanned and imaged at ×20 magnification using the GeoMx DSP with the integrated software suite. Virus-positive and negative ROIs were selected using the polygon segmentation tool with reference to the RNAscope, 161 ROIs were selected in total. Images were then used to identify multiple ROIs on which the instrument focuses UV light (385 nm) to cleave the UV-sensitive probes, with the subsequent release of the hybridized barcodes, and the subsequent aspirate collected into a 96-well DSP collection plate.

**Library preparation, sequencing and data analysis.** Libraries were prepared using GeoMx Seq Code primers (NanoString) and 1× PCR Master Mix (NanoString) and AMPure XP purification. Library quality was checked using an Agilent Bioanalyzer. The libraries were run on an Illumina NovaSeq sequencing system (GeneWiz/Azenta). The FASTQ files from sequenced samples were converted into Digital Count Conversion (DCC) files using the GeoMx NGS pipeline on NanoString's DND platform. The DCC files were uploaded onto the GeoMx DSP analysis suite (NanoString Version 3), where they underwent quality control, filtering, and Q3 normalization[100]. Normalized GeoMx data was analyzed using R base functions and packages, including Searchlight2[99]. ROI of the same type from the same mice were combined for the analysis.

## Immunohistochemistry

Samples were prepared for immunohistochemistry as described for RNAscope. De-paraffinized and rehydrated sections were incubated in 2% hydrogen peroxide/methanol for 15 minutes. Antigen retrieval (pH 7 or 9) was performed using Vector antigen unmasking (low pH) solution (Vector cat #H-3300) and pH 9 (#H-3301) at 100 °C for 20 minutes. Sections were blocked with an avidin/biotin kit (SP-2001; Vector Laboratories, Burlingame, CA) for 20 minutes, then with 10% normal goat serum (NGS, Invitrogen) in PBS for 20 minutes. Slides were incubated overnight at 4 °C with primary antibodies at 1:200 directed against B220 (Invitrogen 14-0452-82, clone; RA36A3). Slides were washed with PBS and then incubated with biotinylated goat anti-rat IgG (ThermoFisher, 31830) for 1.5 hours. Amplification of the antigen was achieved using an R.T.U. Vectastain kit (Vector, PK-7100), and positive cells were visualized by 3,3-diaminobenzidine tetrahydrochloride (Vector, SK-4100). Hematoxylin and Eosin staining was performed to stain nucleic acids and cytoplasmic or extracellular proteins, respectively[17].

## Immunofluorescence

Mice were euthanized by rising concentrations of carbon dioxide ($CO_2$), and the vena cava was cut. Lungs were perfused with PBS-5 mM

ethylenediaminetetraacetic acid to remove red blood cells, and 1% paraformaldehyde was used to fix the lungs. Lung inflation was achieved using 1–3 mL 1% warm UtraPure low-melting agarose administered via the trachea. Lungs with solidified agarose were incubated in 1% paraformaldehyde overnight at 4 °C followed by incubation in 30% sucrose for a further 2–5 days at 4 °C. Lung lobes were frozen in optimal cutting temperature (Tissue-Tek 4583, UK) and stored at −80 °C. Lungs were sectioned into 10μm slices on a Shandon Cryotome FE (Thermo Scientific 12087159, UK) and mounted onto Super Frost slides (Thermo Scientific). Slides were fixed in 100% cold acetone and stored at −20 °C.

Slides were rehydrated in PBS containing 0.5% bovine serum albumin (BSA) for 5 minutes and incubated with Fc block (24G2) for 30–60 minutes at room temperature. The sections were stained with antibodies at 4 °C overnight: anti-CD4 AlexaFluor647 (BD UK, RM4-5), anti-MHCII eFluor 450 (M5114, ThermoFisher, UK), anti-B220-PE (ThermoFisher, RA3-6B2). Slides were washed in PBS-BSA and mounted with VectorShied (Vector, UK). Images were collected on an LSM880 (Zeiss, Germany) confocal microscope at ×20 magnification using ZEN Black (Version 2.3).

### Digital slide scanning
Excised lung tissues for histological analysis were fixed in 10% neutral buffered formalin for 24 hours and paraffin-embedded in a tissue processor (UK, Shandon Pathcentre Tissue Processor). Formalin-fixed and paraffin-embedded tissues were sectioned (6 μm) and processed for subsequent staining with H&E using standard protocols. Stained slides were scanned on a Leica Aperio VERSA 8 bright-field slide scanner (Leica Biosystems, UK) at ×10 or ×20 magnification and were visualized/analyzed using Aperio ImageScope version 12.1.0.5029 (Aperio Technologies Inc., Vista, USA). Immune cell aggregates indicated by dense areas for blue hematoxylin staining of nuclei were identified and quantified by a blinded observer. The number of aggregates present per lung was calculated on a per slide basis. These structures are described in more detail in Hargrave et al.[17]. X31-IAV positive areas detected using ISH were measured using Aperio V9 algorithm (Aperio ScanScope XT System; Aperio Technologies).

### In vitro structural cell infection and T cell interaction assay
CD45-negative cells were isolated from naive mice or those that had been challenged with IAV-WSN 30 days previously, as per flow cytometry tissue preparation above and CD45+ cells were removed using CD45-microbeads as per manufacturer's instructions (Miltenyi-Biotec). Cells were seeded at $1 \times 10^6$ cells per well of a 48-well plate and allowed to adhere overnight. Approximately 18 hours later, cells were infected with IAV-X31 (Multiplicity of Infection (MOI) 1) or PR8, PR8-OVAI, PR8-OVAII at an MOI of 2. T cells were isolated from the spleens of mice that had been infected with IAV-X31 9 days previously to generate a pool of IAV-specific T cells using a T-cell isolation kit (Stem Cell, as per manufacturer's instructions). T cells were cultured with infected and control CD45-negative cells for 24 h. Alternatively, lymph nodes were taken from naive OTI and OTII mice and CD8 T cells from OTI mice were isolated by a negative CD8 isolation kit (Stem Cell, as per manufacturer's instructions). $2 \times 10^5$ OTI and $4 \times 10^5$ OTII cells were added per well, and T cells were analyzed 3 days later. Flow cytometry analysis was performed as described above.

### Luminex assay
The culture supernatants from the in vitro infected stromal cells were harvested, centrifuged to remove cellular debris, and stored at −80 °C until assayed by Luminex (R&D). Each experimental condition was carried out in duplicate. Chemokine/cytokine levels in supernatants were analyzed by quantification with the Milliplex MCYTOMAG-70K assay; IL1α, IL1β, IL10, RANTES/ CCL5, MIG/ CXCL9, IP10/CXCL10, IL6, IL-15, IL-13, IL17A, IL12p70, IL12p40, GRO/CXCL1 and Milliplex MTH17MAG-47K-03 assay; IL33, IL17F, IL22 (EMD Millipore) according to the manufacturer's instructions.

### Statistical analysis
Data were analyzed using GraphPad Prism version 10.01 for Windows (GraphPad). Data were tested for normality using the Shapiro–Wilk test with ($\alpha = 0.05$). Data are presented as mean ± SEM or median ± IQR depending on distribution, and $p$ values were calculated as indicated in the figure legends. $P$ values $< 0.05$ were considered statistically significant.

### Reporting summary
Further information on research design is available in the Nature Portfolio Reporting Summary linked to this article.

## Data availability
The RNAseq data generated in this study have been deposited in the GEO database under accession code GSE278132: Nanostring and GeoMx data are included in Supplementary data file, Source Data are provided with this paper and all data available on request to the corresponding authors. Source data are provided with this paper.

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

## Acknowledgements

We thank the staff of the School of Infection and Immunity Flow Cytometry Facility, Glasgow Imaging Facility, Biological Services and Polyomics, and the team of the Veterinary Histopathology Service at the University of Glasgow for their technical assistance. We acknowledge the laboratory of Professor Alain Townsend at the University of Oxford for providing the SFLU virus, and we would like to thank Dr. Tiong Tan for helpful discussion. OTII mice were provided by Professor James Brewer at the University of Glasgow and OTI mice and PR8, PR8-OVAI, and PR8-OVAII were provided by Dr Edward Roberts at the Cancer Research UK Scotland Institute. The schematics in Fig. 8B and Supplementary Fig. 12A were created in BioRender. Worrell, J. (2025) https://biorender.com/hwavd50; https://BioRender.com/ux9t49r. This work was supported by the Wellcome Trust (210703/Z/18/Z, awarded to M.K.L.M., and 226141/Z/22/Z awarded to MPalmarini); a Rosetrees' Trust Seedcorn Grant (Seedcorn2020\100017, awarded to J.C.W.), a Medical Research Council (MRC) Harwell GEMM grant, (GEMM8 2317, awarded jointly to J.C.W. and M.K.L.M), and MRC (MC_UU_00034/4, awarded to MPalmarini) Wellcome Trust Institutional Strategic Support Funds (097821/Z/11/B awarded to C.H. and M.K.L.M. and 204820/Z/16/Z awarded to J.C.W.), and a University of Glasgow PhD scholarship to G.E.F.

## Author contributions

Study conception: J.C.W., C.H., M.K.L.M.; investigation: J.C.W., K.E.H., G.E.F., C.H., M.Pingen, T.P., K.M., E.B., C.M.D., J.A., G.I., V.H., C.K-D., Y.D.; formal analysis: J.C.W., K.E.H, G.E.F., C.H., J.C., J.S.N. F.M., M.Pingen, E.B., J.A., G.I., V.H., C.K-D., Y.D., M.K.L.M.; data curation: J.C.W., K.E.H., G.E.F., J.C., M.K.L.M.; visualization: J.C.W., M.K.L.M.; funding acquisition: J.C.W., C.H., M.Palmarini, M.K.L.M.; project administration: J.C.W.; M.K.L.M.; supervision: J.C.W., N.B.J, M.Palmarini, M.K.L.M.; writing original draft: J.C.W., M.K.L.M. All authors reviewed and edited the manuscript.

## Competing interests

The authors declare no competing interests.
