## [Peer Review file · Nature Communications]

Lung structural cells are altered by influenza virus leading to rapid immune protection following re-challenge

Corresponding Author: Dr Megan MacLeod

Version 0:

Reviewer comments:

Reviewer #1

(Remarks to the Author)

This study focuses on how epithelial cells, fibroblasts and endothelial cells, collectively referred to in the paper as structural cells, respond following influenza A virus infection, and during a recall infection with a different viral strain. Bulk RNA sequencing is used to show that there are transcriptomic changes that occur ten days after viral infection in all “structural” cell types, and that ~5-10% of these changes persist forty days after infection. Many of these genes are related to antigen presentation, and are SpiB targets. SpiB expression was confirmed in the lung forty days after infection, in areas of the lung that still contained immune cell infiltrates. The presence of SpiB+ cells required viral replication beyond one cell division, linking its expression with ongoing inflammation. SpiB+ structural cells tend to have higher levels of MHCII expression, the effect was strongest at d10, but persisted in some cells until forty days post infection leading to the hypothesis that there may be alterations in response to secondary infection.

The response of structural cells infection was tested in a model in which mice were first infected with WSN (H1N1 influenza A) then infected with HKx31 (H3N2). Compared with primary HKx31 infection, epithelial cells with previous exposure to WSN has less influenza NP protein detectable intracellularly and a reduced interferon response, indicating previous infection with an different influenza A virus limited infectivity. RNAscope for viral RNA combined with GeoMx spatial transcriptomics showed that prior WSN infection limited the spread of inflammation, and increased expression of genes associated with antigen presentation, and response to interferon. This was also associated with lower viral titre, that was maintained with CD4 and CD8 cell depletion. In this system it would also be predicted that cross reactive antibodies would not form because of the different surface viral proteins. In an in vitro HKx31 infection system, epithelial cells that have in vivo exposure to WSN had reduced intracellular NP expression, and increased chemokine and cytokine expression. Overall, this is a nice contribution to the field because it shows that non-immune cells contribute to influenza A virus infections, and I'm supportive of its publication in Nat Comms. The contribution of non-immune cells in viral reinfection is poorly understood, and this study moves the field forward. There are a few points that should be addressed prior to publication.

Points for consideration:

The conclusions, and possibly the data, on NP-staining in fibroblasts should be removed in Figure 7D. Unlike the staining in epithelial cells where this is a discrete population of positive cells, the staining in fibroblasts seems to be because part of a population present in uninfected controls moves into the gate, it is not convincing that it is real. If the authors prefer not to remove this data, then another method should be used to provide more convincing data on the expression in fibroblasts.

The conclusion in the abstract that “These data suggest lung structural cells could be effective targets for vaccines to boost durable protective immunity” is probably an overstatement based on the data. The data in Figure 3 indicates that ongoing lung inflammation may be required to mediate some of the described effects, which would not be a favourable outcome of a vaccine. Likewise, the association in the discussion with “trained immunity” seems also be an overstatement for the same reason, the changes may be due to ongoing inflammation rather than epigenetic rewiring that underpins trained immunity.

This conclusion “This demonstrates that the structural cells could process and present IAV antigens to CD4 and CD8 T cells (Fig 7C and SFig 12A-B)” is an overstatement based on the data. The data show upregulation of activation markers on T cells that are isolated from the spleens of IAV infected mice. Whilst it's possible that this is dependent on antigen presentation, this has not been shown. It's possible that this could be a reaction to the other molecules produced by these

cells, for example those in 7E. If the authors wish to make conclusions about presentation driving the increased activation in T cells this would need to be formally demonstrated.

Reviewer #2

(Remarks to the Author)

In this manuscript, the authors employ transcriptomic and flow cytometry analyses to investigate long-term changes in lung structural cells following primary influenza infection and their subsequent responses to re-challenge with influenza. Their data reveal that prior influenza infection imprints epithelial cells, fibroblasts, and endothelial cells to exhibit increased antigen processing and presentation, as well as enhanced resistance to subsequent viral infection. The main observation of the manuscript is of interest, as there are sparse studies dedicated to investigating this phenomenon. Nevertheless, the work remains mainly descriptive, largely outlining a phenomenon that could be predicted based on several prior studies (though the authors do provide a more comprehensive characterization in this study). In its current form, the study fails to provide mechanistic insights into the processes inducing persistent changes in structural cells and/or their maintenance.

Other Major Specific Comments:

1. Lung epithelial cells (EpCAM+) have many different cell types. The bulk sequencing results might reflect variations in the composition of these epithelial cells rather than changes within specific cell types.
2. The sustained expression of MHC II is intriguing. What are the consequences of the loss of MHC II expression in epithelial cells?
3. It is unexpected that CD4/CD8 depletion did not alter viral titers in Figure 7a, which contradicts the current understanding of memory T cells and influenza protection. Additionally, it is unusual to observe a large number of TCR-β+ cells remaining after CD4 and CD8 depletion.

Minor Comments:

- The statement "surfactant-producing club epithelial cells" in the introduction is inaccurate, as surfactant is mainly produced by alveolar type II (ATII) cells.

Reviewer #3

(Remarks to the Author)

In this study, Worrell et al. provided an in-depth characterization of the phenotypic changes induced by Influenza A Virus (IAV) in lung structural cells. They employed multiple technical approaches, including transcriptomics, flow cytometry, and immunofluorescence, to investigate these long-term changes and how they affect the antiviral response upon re-challenges. While the general idea of the work is clear, and structural cells are increasingly recognized as targets for novel vaccine development, many important conclusions regarding the role of epithelial cells in antigen presentation and communication with T cells are not conclusively demonstrated. Therefore, despite the potential interest, we consider that the novelty of the work is limited, and we recommend publication in another journal.

General comments.

- In Fig. 3A, the authors show the SpiB target genes, but the expression levels of SpiB do not seem to align with the qPCR data presented in SFig 4B, which indicates that SpiB expression was significantly higher at d40 in both epithelial and stromal cells.
- In Fig. 3C, they claim that in infected animals with no inflammation, SpiB expression is absent. However, they do not perform counterstaining for the virus in the sections labeled as "no inflammation," leaving uncertainty as to whether these lungs were properly infected. In addition, the detection of SpiB by RNAscope analysis appears quite faint. Since they have SpiB-mCherry animals, it might be interesting to improve this analysis using these animals. Additionally, it would be beneficial to include day 10 sections as controls since they claim SpiB is associated with both inflammation and viral replication.
- In Fig. 4, there is no evident increase in the SpiB+ population at day 40. How do you explain the increase in RNA levels seen in SFig 4B? To better address this, it may be helpful to correlate transcriptomic data with protein levels through Western Blot analysis.
- In Fig. 4, the authors emphasize the role of alveolar type II (ATII) cells, which are the only structural cells expressing SpiB in IAV presentation, by analyzing MHCII. Although these findings are critical to the paper, we find them not very convincing. The expression of MHCII by SpiB+ ATII cells is not increased compared to SpiB- cells. Furthermore, the basal levels before infection are not shown. To reinforce this part of the study, the authors could use SpiB knockout (KO) animals to determine whether the absence of this protein affects MHCII expression upon infection.
- Fig. 5 lacks a neutralization assay demonstrating that there is no cross-protection between mice infected with WSN and X31. They should include data showing that serum from WSN-infected animals cannot neutralize the X31 virus. This would strengthen the findings regarding IAV re-challenge.
- Figs. 6 and 7 highlight the role of epithelial cells in IAV presentation and T-cell communication; however, many aspects are properly addressed. In Fig. 6D, the authors indicate that there is still inflammation upon re-challenge, but this is not type I and is not associated with viral replication. They observed increased IL-1α in vitro and improved type II IFN in vivo. Is this new inflammatory response beneficial for enhancing viral clearance? How about other immune cells, such as antigen-presenting cells (APCs) or natural killer (NK) cells, which might respond to IL-1α and secrete type II IFN? We suggest investigating these additional cell types to clarify the link between stromal cells and T cells.
- In Fig. 7C, they conclude that the slightly lower expression of CD25 and CD69 correlates with a memory phenotype of stromal cells, suggesting that these cells are better at presenting antigens. However, to sustain the claim of antigen presentation by these cells, it would be beneficial to analyze the proliferation of HA-TCR-specific CD4 and CD8 T cells after co-culturing with influenza-infected stromal cells.

Minor:

- In SFig 3, the correlation between RNA-seq and nCounter analyses omits data for endothelial cells.
- In SFig 7, they utilized flow cytometry to improve the resolution of SpiB+ cells. However, they could have employed immunofluorescence of SpiB-mCherry+ lungs, which would also provide spatial co-localization of inflamed EpCAM+ and SpiB+ cells.
- Fig. 5A lacks a control group showing the number of EpCAM+ NP cells in WSN-infected animals.
- In Fig. 5D, they claim that fibroblasts are less susceptible to infection, as reported by Boyd et al. in Nature 2020. However, they should provide the number of fibroblast NP+ cells to support this claim.

Reviewer #4

(Remarks to the Author)

Version 1:

Reviewer comments:

Reviewer #1

(Remarks to the Author)

The authors have been thorough in their review, and I congratulate them on a nice paper.

Reviewer #2

(Remarks to the Author)

Unfortunately, the authors did not adequately address my critical concerns in the revised manuscript. For instance, epithelial cell types are challenging to identify accurately by flow cytometry. This raised concerns about the validity of the new data presented in Figure 4. Additionally, the authors' argument regarding the role of memory T cells in protection against infection appears unconvincing. Numerous studies have already demonstrated the essential role of memory T cells, particularly tissue-resident memory T cells, in limiting early viral replication, including during influenza virus infection. The claim that "most studies examine viral titres at day 5 or 6 after re-infection" does not accurately reflect the current literature. Without compelling evidence to challenge the prevailing understanding, the rigor and robustness of the data presented in the manuscript remain questionable.

Reviewer #3

(Remarks to the Author)

We do appreciate the effort that the authors have put in the revision process and we consider that the major points raised have been properly address. We think that the new version of the manuscript has significantly improved. Therefore, we are positive regarding publication of this work in its current form in nature communications.

We thank the reviewers for taking time to review our manuscript and for their helpful comments. We would like to respond to them in the order they were presented:

Reviewer 1: Remarks to authors

This study focuses on how epithelial cells, fibroblasts and endothelial cells, collectively referred to in the paper as structural cells, respond following influenza A virus infection, and during a recall infection with a different viral strain. Bulk RNA sequencing is used to show that there are transcriptomic changes that occur ten days after viral infection in all “structural” cell types, and that ~5-10% of these changes persist forty days after infection. Many of these genes are related to antigen presentation, and are SpiB targets. SpiB expression was confirmed in the lung forty days after infection, in areas of the lung that still contained immune cell infiltrates. The presence of SpiB+ cells required viral replication beyond one cell division, linking its expression with ongoing inflammation. SpiB+ structural cells tend to have higher levels of MHCII expression, the effect was strongest at d10, but persisted in some cells until forty days post infection leading to the hypothesis that there may be alterations in response to secondary infection.

The response of structural cells infection was tested in a model in which mice were first infected with WSN (H1N1 influenza A) the infected with HKx31 (H3N2). Compared with primary HKx31 infection, epithelial cells with previous exposure to WSN has less influenza NP protein detectable intracellularly and a reduced interferon response, indicating previous infection with an different influenza A virus limited infectivity. RNAscope for viral RNA combined with GeoMx spatial transcriptomics showed that prior WSN infection limited the spread of inflammation, and increased expression of genes associated with antigen presentation, and response to interferon. This was also associated with lower viral titre, that was maintained with CD4 and CD8 cell depletion. In this system it would also be predicted that cross reactive antibodies would not form because of the different surface viral proteins. In an in vitro HKx31 infection system, epithelial cells that have in vivo exposure to WSN had reduced intracellular NP expression, and increased chemokine and cytokine expression. Overall, this is a nice contribution to the field because it shows that non-immune cells contribute to influenza A virus infections, and I'm supportive of its publication in Nat Comms. The contribution of non-immune cells in viral reinfection is poorly understood, and this study moves the field forward. There are a few points that should be addressed prior to publication.

Reviewer 1: Response to remarks

We thank R1 for their positive comments and succinct summary of our manuscript.

Reviewer 1: Comment 1

Points for consideration: The conclusions, and possibly the data, on NP-staining in fibroblasts should be removed in Figure 7D. unlike the staining in epithelial cells where this is a discrete population of positive cells, the staining in fibroblasts seems to be because part of a population present in uninfected controls moves into the gate, it is not convincing that it is real. If the authors prefer not

to remove this data, then another method should be used to provide more convincing data on the expression in fibroblasts.

Reviewer 1: Response 1:

First, we want to provide some more detail on the methodology in this experiment:

Lung CD45 negative cells from each lung were infected in duplicate *in vitro* to provide technical replicates. In addition to the biological replicates, this provided us with greater confidence in the data. To set the gates for NP+ cells, we used the uninfected control for each cell type to account for any differences in autofluorescence and/or non-specific binding of the anti-IAV NP antibody. For the graphed data, we used the mean of the two technical duplicates for each biological replicate and picked a representative FACS plot that was around the mean of the group to display in the paper.

We have provided additional flow cytometry data showing IAV-NP staining in lung fibroblasts as REVIEW ONLY material. This includes the corresponding replicate data (Review only Figure 1A) for the original FACS plots included in original Figure 7D (now Figure 8D) of the manuscript. We have also included IAV-NP staining from a separate independent experiment showing increased IAV-NP staining corresponds with increasing MOI of X31 virus infection *in vitro* (Review only Figure 1B).

The IAV-NP antibody was also used to stain fibroblasts in whole lung issue digests at day 2 post *in vivo* IAV infection (see Reviewer 3, Minor comment 3, Response 4).

We hope that the additional data provided convinces the Reviewer that the fibroblast IAV-NP staining is real.

Reviewer 1: Comment 2:

The conclusion in the abstract that “These data suggest lung structural cells could be effective targets for vaccines to boost durable protective immunity” is probably an overstatement based on the data. The data in Figure 3 indicates that ongoing lung inflammation may be required to mediate some of the described effects, which would not be a favourable outcome of a vaccine. Likewise, the association in the discussion with “trained immunity” seems also be an overstatement for the same reason, the changes may be due to ongoing inflammation rather than epigenetic rewiring that underpins trained immunity.

Reviewer 1: Response 2:

At the suggestion of Reviewer 1, we have modified our conclusions on vaccine outcomes and trained immunity in the current version of the manuscript abstract.

We have removed the sentence:

“These data suggest lung structural cells could be effective targets for vaccines to boost durable protective immunity”

We have replaced it with:

“These data suggest lung structural cells display characteristics of immune memory which could affect outcomes that are protective in the context of infection or pathogenic in chronic inflammatory disorders.”

We acknowledge the limitations of our study in relation to trained immunity and have added the following text to the discussion:

“We did not investigate epigenetic modifications to the structural cells, this is a known hallmark of trained immunity. Therefore, it is not possible to determine if the changes we observed are due to ongoing inflammation, potentially caused by ongoing antigen presentation, or the epigenetic rewiring that underpins trained immunity.”

Reviewer 1: Comment 3:

This conclusion “This demonstrates that the structural cells could process and present IAV antigens to CD4 and CD8 T cells (Fig 7C and SFig 12A-B)” is an overstatement based on the data. The data show upregulation of activation markers on T cells that are isolated from the spleens of IAV infected mice. Whilst it’s possible that this is dependent on antigen presentation, this has not been shown. It’s possible that this could be a reaction to the other molecules produced by these cells, for example those in 7E. If the authors wish to make conclusions about presentation driving the increased activation in T cells this would need to be formally demonstrated.

Reviewer 1: Response 3:

We have now formally demonstrated that lung structural cells can present antigen to T cells and have included these data in the manuscript. The results are shown in new Supp Fig 14.

To do this, we used IAV-PR8 viruses that contain Ovalbumin epitopes, either SIINFEKL recognised by OTI T cell receptor (TCR) transgenic (Tg) CD8 T cells, or OVA-323-339, recognised by OTII TCR Tg CD4 T cells. We infected lung CD45 negative cells from either naïve mice or mice infected 73 days earlier with IAV-X31 with these viruses or control wild-type PR8 *in vitro*. The T cells were added after 24 hours and T cell activation examined after a further three days.

These data show clear CD25 and CD69 expression in co-cultures with the relevant antigen, e.g. OTII cells only upregulate these markers in co-cultures infected with PR8 expressing OVA323-339. We did see a significant although very slight increase in CD69 expression by OTI T cells in co-cultures with PR8 expressing OVA323-339. This minor upregulation of CD69 may be driven by the activation of the OTII cells in the same co-culture, as this was not observed in the co-culture with PR8 infected cells.

These data are included as new Supplementary Figure 14 and together with Fig 8C (previous Fig7C), show that the CD45 negative cells can present antigen to T cells.

Reviewer 2: Remarks to authors

In this manuscript, the authors employ transcriptomic and flow cytometry analyses to investigate long-term changes in lung structural cells following primary influenza infection and their subsequent responses to re-challenge with influenza. Their data reveal that prior influenza infection imprints epithelial cells, fibroblasts, and endothelial cells to exhibit increased antigen processing and presentation, as well as enhanced resistance to subsequent viral infection. The main observation of the manuscript is of interest, as there are sparse studies dedicated to investigating this phenomenon. Nevertheless, the work remains mainly descriptive, largely outlining a phenomenon that could be predicted based on several prior studies (though the authors do provide a more comprehensive characterization in this study). In its current form, the study fails to provide

mechanistic insights into the processes inducing persistent changes in structural cells and/or their maintenance.

Reviewer 2: Response to remarks:

Thank you to the reviewer for noting that our manuscript addresses an area of research in which there are few studies. We would like to take this opportunity to highlight some key novel findings from this manuscript, including new data we have added, to make the impact of our findings clearer. We also direct the reviewer to the Discussion where we explain how our findings advance the field of epithelial cells, fibroblasts, and anti-viral immunity in general.

1. The paradigms of immunology are shifting: The immune-cell centric focus is giving way to an understanding that many different cell types contribute to long term protective immunity. Our data provide a detailed assessment of the immune-related longer-term consequences of influenza virus infection on lung structural cells. Our study, therefore, is a critical step-forward providing a more in-depth understanding of tissue-level changes following infection than current studies.
2. Our study has breadth across different lung structural cells, but we focus in on epithelial cells as they are the first infected with influenza virus. Published studies have highlighted the diversity of epithelial cell subsets mostly *via* imaging and sequencing studies. We have combined our findings from the RNA sequencing data with flow cytometry to generate robust quantifiable data on multiple different epithelial cell types. Given the key role of T cells in virus clearance following primary infection and heterosubtypic challenge, we examined the expression of MHCI (new data) and MHCII on mature and progenitor epithelial cell populations.

Our data demonstrate increased MHCI and MHCII expression in mature (ciliated and club) and progenitor (Sca1+) populations following influenza virus infection. While others (e.g. Shenoy *et al* PMID: 34611166) have described increased MHCII expression following infection, our long-term data are the first to show sustained increased expression of MHCI and MHCII at protein level in the lung following infection.

These data have two key implications: First, defining which epithelial cells are more likely to interact with T cells within the lung, potentially supporting protective (e.g. anti-pathogen) or pathogenic (e.g. allergic) responses. Second, the presence of altered MHCI and MHCII protein expression in a progenitor population indicates that the memory of the infection has the potential to outlive the lifespan of the mature epithelial cell populations.

Additionally, to increase the granularity of our understanding of epithelial cell diversity, we generated a new reporter mouse for the transcription factor SpiB. Our data reveal that SpiB+ epithelial cells are more likely to display persistent changes in MHCI/II expression than SpiB negative cells. The reporter mice are therefore a useful tool to explore epithelial cell memory in the future.

3. In addition to demonstrating long-term changes at the level of gene transcription and protein expression, we show that lung epithelial cells from previously infected mice can control virus more effectively. This functional intrinsic immune memory in epithelial cells

represents an exciting shift in the paradigm that immune memory is hallmark of adaptive, or any other, immune cells.

This more rapid virus control was accompanied by an increased cytokine response, strengthening our overall conclusion that prior infection leads to sustained changes in lung structural cells.

4. Our analysis is the first to address how prior infection leads to an altered response by IAV infected epithelial cells. Our spatial transcriptomic data demonstrate an altered type of response and altered spatial distribution of the anti-viral response. These data provide a potential explanation for reduced symptoms in re-infected individuals that goes beyond rapid virus clearance.

In the time available for the review, we have not been able to generate data that address underpinning mechanism. We argue that the current manuscript contains exciting, robust and novel findings that are central to furthering the fields acceptance that functional immune memory exists beyond traditional immune cells.

Reviewer 2: Comment 1:

Lung epithelial cells (EpCAM+) have many different cell types. The bulk sequencing results might reflect variations in the composition of these epithelial cells rather than changes within specific cell types.

Reviewer 2: Response 2

We agree that the bulk RNA-sequencing studies contain heterogenous populations of epithelial cells. We use these data as a starting point to investigate long-term changes in the lung structural cells. We chose to mainly focus our follow up studies on epithelial cells as these cells are first responders to influenza virus infection. We did these experiments using flow cytometry which provides robust quantifiable data at a protein level.

We conducted detailed flow cytometry analysis on wild type C57BL/6 mice and found no changes in epithelial cell subset at baseline (day 0) compared to the memory timepoint. We examined cell numbers and proportions of ciliated, club, progenitor and alveolar epithelial cells. These experiments were performed in the same manner as the SpiB reporter studies that are included in Fig 4.

We have not included the wildtype C57BL/6 data in the current manuscript as the SpiB data provides the same information and additional detail. It does give us added confidence that the transcriptional changes to EpCAM1+ cells are unlikely to be driven by altered epithelial cell subset composition and are instead due to intrinsic transcriptional alterations to the epithelial cells.

The SpiB flow cytometry analysis enabled us to demonstrate short and long-term changes in MHCI and MHCII expression by lung EpCAM1+ epithelial cells. Briefly, IAV infection led to upregulation of MHCI and MHCII in all epithelial cells (Fig 4-5 and new SFig9). This upregulation is more substantial in SpiB+ cells compared to SpiB negative cells. Furthermore, the percentage of cells that are MHCI/II high remain elevated in SpiB+ ciliated, club and Sca1+ progenitor cells at day 30, suggesting that memory is held by both long lived populations and at the stem cell level (Fig 5B). This is consistent

with previous findings in 'long lived epithelial cells': club cells, PMID: 27001854; Ciliated cells PMID: 31557273]. Our data go beyond these studies by defining an altered ability of these cells to communicate with T cells.

See also Reviewer 3 Response 4.

Reviewer 2: Comment 2

The sustained expression of MHC II is intriguing. What are the consequences of the loss of MHC II expression in epithelial cells?

Reviewer 2: Response 2

There are some recent studies using cell conditional knockout of MHCII in lung epithelial subsets that have investigated some aspects of this question.

A study by Toulmin *et al.* [PMID: 34183650] used mice with an ATII specific depletion of MHCII (SPC^{ΔAb1}). Their main finding was that loss of MHCII on ATII cells led to increased weight loss following Sendai virus infection. SPC^{ΔAb1} animals also showed an elevated, but not statistically significant mortality rate following IAV infection.

Shenoy *et al.*, [PMID: 34611166] demonstrated that loss of MHCII on all lung epithelial cells led to a reduction of CD4 memory T cells at the bronchiole-vascular border and small changes in the proportion of Th1 and Th17 following bacterial infection.

We agree that it would be interesting to examine the consequences of loss of MHCII in lung epithelial cells in secondary responses. To perform this study we would require an epithelial cell type conditional knockout that could be controlled temporally, e.g. Cre-ERT2 expressed by only ciliated, club or Sca1 progenitor cells. These studies were beyond the timeline provided for the review and beyond the funding we have available for this study.

Reviewer 2: Comment 3

It is unexpected that CD4/CD8 depletion did not alter viral titers in Figure 7a, which contradicts the current understanding of memory T cells and influenza protection. Additionally, it is unusual to observe a large number of TCR-β+ cells remaining after CD4 and CD8 depletion.

Reviewer 2: Response 3

The reviewer is correct that the current consensus in the field is that CD4 and CD8 T cells mediate heterosubtypic immunity (e.g. reviewed in Gruta and Turner, 2014, PMID: 25043801). There is also some evidence that antibodies and B cells can play a role, although likely by enhancing CD8 T cell responses and working with CD4 T cells to produce new neutralising antibody (Rangel-Moreno *et al.*, 2008, PMID: 19097047).

Most studies examine viral titres at day 5 or 6 after re-infection or measure weight loss over a period of about one week. These data clearly show a requirement for T cells in controlling IAV re-infections. The exception is Epstein *et al.*, 1997, (PMID: 9013963) who examined viral titres at a number of time points following the challenge infection (Table IV, Epstein). In comparison to control animals, there

was a reduction in lung virus titre as early as day 2 post-infection in the absence of CD4 and CD8 T cell). We referenced Epstein *et al* in the original manuscript.

We addressed the requirement for T cells in protective responses to repeat IAV infection in the discussion:

“Our data show that early protection to re-infection was independent of T cells, but using plaque assays and RNAscope, virus is still present at day 2 following the re-challenge infection. Both CD4 and CD8 T cells protect mice and humans from IAV re-infection, especially when priming and re-challenge strains express different hemagglutinin and neuraminidase proteins^{18,20,21,29,30,80,81}. The sustained expression of inflammatory chemokines and molecules involved in the processing and presentation of antigen following infection suggests that lung structural cells will have an enhanced ability to attract and communicate with local T cells. In summary, our data suggests that lung epithelial cells may play both early T cell independent and then later T cell dependent roles in accelerating viral clearance following re-infection.”

In terms of the remaining TCR+ cells in the CD4/8 antibody treated mice, we thank the reviewer for pointing this out. We returned to the original flow data and re-gated the cells. We had MHCII in this panel and we examined MHCII expression alongside TCR expression. This allowed us to identify TCR+ MHCII+, cells which may be activated gamma-delta T cells (e.g. see Cha *et al*, PMID: 32582168 & Cheng *et al*, PMID: 18774183).

We have updated our gating and re-calculated the percentages and numbers of CD4 and CD8 T cells in Supplemental Figure 12. As in the previous gating this shows a substantial loss of CD4 and CD8 T cells.

In the manuscript, our point is not that epithelial cells in mice previously infected with IAV are the ONLY cell type responsible for the early virus control at day 2, but that these cells are able to control virus more effectively, which we tested in the much more controllable *in vitro* setting.

We have added the additional sentences to the discussion to make this clearer:

“It is possible that multiple cell types can contribute to the reduced virus titre at day 2 following infection, these could include altered alveolar macrophages^{82,83} and/or $\gamma\delta$ T cells⁸⁴. We focussed on structural cells as they, in particular epithelial cells, are the first cells infected with virus and thus the first to make an anti-viral response. The reduction of virus only in the epithelial cells in the *in vitro* model indicates cell intrinsic protective response in these cells.”

Minor Comments:

The statement "surfactant-producing club epithelial cells" in the introduction is inaccurate, as surfactant is mainly produced by alveolar type II (ATII) cells.

Response:

This sentence has been corrected and now reads:

“In the context of IAV, club epithelial cells that survive IAV infection can produce a rapid anti-viral response following a second challenge which may contribute to tissue damage”

Reviewer 3: Remarks to authors

In this study, Worrell et al. provided an in-depth characterization of the phenotypic changes induced

by Influenza A Virus (IAV) in lung structural cells. They employed multiple technical approaches, including transcriptomics, flow cytometry, and immunofluorescence, to investigate these long-term changes and how they affect the antiviral response upon re-challenges. While the general idea of the work is clear, and structural cells are increasingly recognized as targets for novel vaccine development, many important conclusions regarding the role of epithelial cells in antigen presentation and communication with T cells are not conclusively demonstrated. Therefore, despite the potential interest, we consider that the novelty of the work is limited, and we recommend publication in another journal.

Reviewer 3: Response to remarks

We would like to emphasize the novelty of data, which we also summarised for Reviewer 2 above:

We hope this summary convinces the reviewer of the importance of our research.

1. The paradigms of immunology are shifting: The immune-cell centric focus is giving way to an understanding that many different cell types contribute to long term protective immunity. Our data provide a detailed assessment of the longer-term immune-related consequences of influenza virus infection on lung structural cells. Our study, therefore, is a critical step-forward providing a more in-depth understanding of tissue-level changes following infection than current studies.
2. Our study has breadth across different lung structural cells, but we focus in on epithelial cells as they are the first infected with influenza virus. Published studies have highlighted the diversity of epithelial cell subsets mostly *via* imaging and sequencing studies. We have combined our findings from the RNA sequencing data with flow cytometry to generate robust quantifiable data on multiple different epithelial cell types. Given the key role of T cells in virus clearance following primary infection and heterosubtypic challenge, we examined the expression of MHCI (new data) and MHCII on mature and progenitor epithelial cell populations.

Our data demonstrate increased MHCI and MHCII expression in mature (ciliated and club) and progenitor (Sca1+) populations following influenza virus infection. While others (e.g. Shenoy *et al* PMID: 34611166) have described increased MHCII expression following infection, our long-term data are the first to show sustained increased expression of MHCI and MHCII at protein level in the lung following infection.

These data have two key implications: First, defining which epithelial cells are more likely to interact with T cells within the lung, potentially supporting protective (e.g. anti-pathogen) or pathogenic (e.g. allergic) responses. Second, the presence of altered MHCI and MHCII protein expression in a progenitor population indicates that the memory of the infection has the potential to outlive the lifespan of the mature epithelial cell populations.

Additionally, to increase the granularity of our understanding of epithelial cell diversity, we generated a new reporter mouse for the transcription factor SpiB. Our data reveal that SpiB+ epithelial cells are more likely to display persistent changes in MHCI/II expression than SpiB negative cells. The reporter mice are therefore a useful tool to explore epithelial cell memory in the future.

3. In addition to demonstrating long-term changes at the level of gene transcription and protein expression, we show that lung epithelial cells from previously infected mice can control virus more effectively. This functional intrinsic immune memory in epithelial cells represents an exciting shift in the paradigm that immune memory is hallmark of adaptive, or any other, immune cells.

This more rapid virus control was accompanied by an increased cytokine response, strengthening our overall conclusion that prior infection leads to sustained changes in lung structural cells.

4. Our analysis is the first to address how prior infection leads to an altered response by IAV infected epithelial cells. Our spatial transcriptomic data demonstrate an altered type of response and altered spatial distribution of the anti-viral response. These data provide a potential explanation for reduced symptoms in re-infected individuals that goes beyond rapid virus clearance.

Reviewer 3: Comment 1

In Fig. 3A, the authors show the *SpiB* target genes, but the expression levels of *SpiB* do not seem to align with the qPCR data presented in SFig 4B, which indicates that *SpiB* expression was significantly higher at d40 in both epithelial and stromal cells.

Reviewer 3: Response 1

In Figure 3A, the RNA sequencing data showing *SpiB* expression in epithelial cells and fibroblasts at day 40 is presented using scaled expression represented by a Z score. It shows elevated *SpiB* expression at day 10 and 40 following IAV compared to expression in naïve animals. In this instance, the z-score represents how much the expression of *SpiB* in a specific sample deviates from the mean expression level of *SpiB* across all samples. The *SpiB* qPCR data presented in SFig 4B was generated using the absolute quantitation method. This measures the exact copy number of *SpiB* in a sample by comparing the Cq (quantification cycle) values of unknown samples to a standard curve generated from known amounts of the *SpiB* target DNA. This method provides a direct measure of the target cDNA concentration.

Therefore, the apparent deviation in *SpiB* expression at day 40 can be explained as the qPCR assay (Supp Fig 4B) is a more direct measure of the number of copies of *SpiB* unlike relative quantification levels (Fig 4A) comparing gene expression levels across all samples in the heatmap (Z-score). Importantly, both methods confirm that *SpiB* is persistently elevated at day 40 post IAV in infection in lung epithelial cells.

Reviewer 3: Comment 2

In Fig. 3C, they claim that in infected animals with no inflammation, *SpiB* expression is absent. However, they do not perform counterstaining for the virus in the sections labeled as “no inflammation,” leaving uncertainty as to whether these lungs were properly infected. In addition, the detection of *SpiB* by RNAscope analysis appears quite faint. Since they have *SpiB*-mCherry animals, it might be interesting to improve this analysis using these animals. Additionally, it would be beneficial to include day 10 sections as controls since they claim *SpiB* is associated with both inflammation and viral replication.

Reviewer 2: Response 2

All mice included in this study are monitored for signs of IAV infection, e.g. weight loss, indicating they have been appropriately infected with virus. However, the IAV infection process is not uniform across all lung lobes and all parts of each lobe, leading to variable levels of infection, and lung injury [PMID: 25807527]. We performed detailed histological analysis of the IAV infected mouse lung in a separate study [PMID: 38851589]. These findings were consistent with other investigations showing certain lung areas display increased damage/inflammation [PMID: 33979629], or foci of infection [PMID: 34739343] compared to other areas.

At the memory timepoint, the virus is no longer present in the lung, and we did not perform counterstaining for the virus. Likewise, most of the virus has been cleared by day 10 post-infection [PMID: 10411921, PMID: 20410284]. We have shown this by flow cytometry analysis using the IAV-NP antibody on the whole lung tissue digests (updated Fig 6A (previous Fig5), *see also Reviewer 3, Minor Response 1*).

The *SpiB* RNAscope staining shown in Fig 3C in the epithelium is faint, however it is present, and it is important to note that this matches our qPCR analysis (SFig 7B) that shows low levels of *SpiB* transcript in cells that report SpiB expression *via* mCherry. We have performed semi-quantitative RNAscope analysis of lung sections from multiple mice, in 2 independent experiments in the manuscript (updated Fig 3C). We have updated Figure 3C to include additional images of *SpiB*+ airway epithelium from an animal in the second experimental cohort.

In Fig 3C, the section labelled as 'no inflammation' and the section labelled as '+ inflammation' come from the lungs of the same animal, stained on the same slide (*SpiB* RNAscope). Reviewer 3 is correct, the tissue area labelled 'no inflammation' was likely an area of the lung in which the virus did not infect any cells. However, this is consistent with our assertion that viral replication/inflammation is required for *SpiB* upregulation in the lung (Fig 3D). To investigate this, we purposely used a virus capable of only one cycle of replication culling mice at the memory timepoint. We think this is a better control to test the role of virus replication than to examine a day 10 time point as the reviewers suggested. This is especially the case as the virus is largely cleared by day 10 following infection.

We have updated the text in the figure legend to read:

"C: Lungs were taken from naïve or C57BL/6 mice infected with IAV 30-40 days previously with representative images from mice from two independent experiments shown. Areas of infected lungs with no inflammation have *SpiB* negative airways while *SpiB* (black arrows) is localized in areas of inflammation (dark red colour, indicated by white star) within the lung and in airways, close to areas of inflammation; adjacent images showing areas of no inflammation and inflammation show lungs from the same mouse."

Reviewer 3: Comment 3

In Fig. 4, there is no evident increase in the SpiB+ population at day 40. How do you explain the increase in RNA levels seen in SFig 4B? To better address this, it may be helpful to correlate transcriptomic data with protein levels through Western Blot analysis.

Reviewer 3: Response 3

SFig 4B shows that *SpiB* is increased using qPCR analysis (absolute quantitation) on sorted EpCAM1+ lung epithelial cells. This assay (SFig 4B) directly measures of the number of copies of *SpiB*. In Figure 4, we used flow cytometry on the *SpiB* reporter mouse to examine protein levels in lung epithelial cell subsets. We found elevated numbers of *SpiB*+ club cells ($p=0.021$) at day 30 (Figure 4C). Therefore, both Fig 4 and SFig 4B show elevated *SpiB* at the memory timepoint using different methods.

Reviewer 3 raises an interesting point regarding *SpiB* protein levels. There are currently no peer reviewed/publication quality antibodies available to measure *SpiB* by Western blot, this is a limitation for the wider research community.

Reviewer 3: Comment 4

In Fig. 4, the authors emphasize the role of alveolar type II (ATII) cells, which are the only structural cells expressing *SpiB* in IAV presentation, by analyzing MHCII. Although these findings are critical to the paper, we find them not very convincing. The expression of MHCII by *SpiB*+ ATII cells is not increased compared to *SpiB*- cells. Furthermore, the basal levels before infection are not shown. To reinforce this part of the study, the authors could use *SpiB* knockout (KO) animals to determine whether the absence of this protein affects MHCII expression upon infection.

Reviewer 3: Response 4

We apologise if Figure 4 was unclear. We have updated the figure legend and manuscript text in the results section to improve the clarity of the paper. We do not agree that we have emphasised ATII cells, which are known to express MHCII (e.g. Toulmin, PMID: 34183650). We summarise below our findings presented in Fig4:

Fig4A shows which types of epithelial cells are found within *SpiB* negative and *SpiB* positive populations. ATII cells are indeed the largest population within both the *SpiB* positive and negative populations. The p values on the graph show whether the indicated population is significantly higher in the *SpiB* negative or positive population. We did not explain this clearly and have added the following sentence to the figure legend:

'The p -values indicate that the population is present at a significantly greater percentage within the *SpiB* negative or positive subset.'

In Fig 4B we display the percentage of *SpiB* neg/pos epithelial cells of each type that are MHCII in naïve animals -these are the baseline levels that the reviewer has requested. These data show that for all the epithelial populations, the *SpiB*+ fractions are more likely to express MHCII than the *SpiB* negative cells.

The original text reads:

'For all the populations, *SpiB*+ cells were more likely to be MHCII+ than *SpiB* negative populations, this included ATII cells that are predominately MHCII+ in which we found that *SpiB*+ cells were slightly, but significantly, more likely to express MHCII (Fig4B, SFig 7D).'

Additionally, SFig 7D shows representative FACS plots for each epithelial cell population.

We have also updated the figure title to emphasise the finding:

'Figure 4: *SpiB*+ epithelial cells express more MHCII than *SpiB* negative cells'

Fig 4D shows that the percentage of ciliated cells that express MHCI is increased at day 10 following infection and remains increased at day 30 in the SpiB+ populations. Similarly, the amount of MHCI (measured by MFI) is increased in club cells, Sca1 progenitor cells and ATII cells at either day 10 or day 30 following infection.

We have now performed additional experiments to analyse MHCI expression by SpiB negative and positive epithelial cells. These data are shown in new SFig 8 and new Fig 5. The key finding from these data is that infection leads to increases in an MHCI-MHCI high population and that in ciliated, club and Sca1+ progenitor populations, this population remains increased at day 30. Of note, at day 30, this population has increased 8, 9 and 13-fold in the SpiB positive ciliated, club and Sca1+ progenitor cells in populations from naïve compared to day 30 mice respectively.

In summary, our data reveal much more complexity in MHCI epithelial cell expression than previously recognised and carefully chart these changes during a primary infection and at memory time points. Our data suggest that several different epithelial cells display long term changes in MHCI and MHCI expression, indicating an increased ability to communicate with both CD4 and CD8 T cells.

The reviewer also asked about experiments in SpiB KO mice. These animals have intrinsic defects in the B cell compartment (Garrett-Sinha *et al*, Immunity 1999, PMID: 10229183), and thus would unlikely experience a normal primary antigen specific immune response to influenza virus. While it would be interesting to remove SpiB from individual epithelial cell populations at a memory influenza time point using cell and temporal conditional knockout animals, such studies are beyond the scope of our study.

Reviewer 3: Comment 5

Fig. 5 lacks a neutralization assay demonstrating that there is no cross-protection between mice infected with WSN and X31. They should include data showing that serum from WSN-infected animals cannot neutralize the X31 virus. This would strengthen the findings regarding IAV re-challenge.

Reviewer 3: Response 5

Using subsequent infections with H1N1 (WSN or PR8) and H3N2 (X31) (or vice versa) is a well-established method to study heterosubtypic immunity to IAV. The Haemagglutinin (H) classification system is based on a lack of antibody recognition to the viral glycoproteins haemagglutinin and neuraminidase (PMID: 39890719) between viruses. There is a lot of interest in broadly neutralising antibodies in the context of IAV, these antibodies often recognise the less variable stalk region of haemagglutinin. However, H1 and H3 are distinct in the stalk region and antibodies do not cross react (Epstein and Price, 2010 PMID: 21087110). This is nicely illustrated by Baker *et al*, 2015, (PMID: 26044496) who have developed an imaging based system to study IAV antibody recognition. Figure 2B (Baker) provides examples of monoclonal or polyclonal Ab binding to H1 and H3 and show that antibodies recognising one haemagglutinin do not bind the other haemagglutinin.

As discussed in the manuscript, the current consensus in the field is that T cells are responsible for IAV heterosubtypic immunity (e.g. reviewed in Gruta and Turner, 2014, PMID: 25043801). These studies are based on analysis of virus at day 5 or 6 after infection, or weight loss of a week or longer. As we note in the manuscript, Epstein *et al* 1997, (PMID: 9013963) also looked at day 2 following a re-infection and found that T cells were not required for enhanced virus control at this time point.

There is some evidence that B cells and non-neutralising antibody can play a role in this protection, but this is likely mediated by enhancing the CD8 T cell responses and by producing new neutralising antibodies, which takes several days and requires CD4 T cell help (Rangel-Moreno et al, 2008, PMID: 19097047).

See also Reviewer 2 Response 3.

Reviewer 3: Comment 6

Figs. 6 and 7 highlight the role of epithelial cells in IAV presentation and T-cell communication; however, many aspects are properly addressed. In Fig. 6D, the authors indicate that there is still inflammation upon re-challenge, but this is not type I and is not associated with viral replication. They observed increased IL-1a in vitro and improved type II IFN in vivo. Is this new inflammatory response beneficial for enhancing viral clearance? How about other immune cells, such as antigen-presenting cells (APCs) or natural killer (NK) cells, which might respond to IL-1a and secrete type II IFN? We suggest investigating these additional cell types to clarify the link between stromal cells and T cells.

Reviewer 3: Response 6

Thank you to the reviewers for their suggestions for additional studies and an opportunity to describe in more detail previous studies from our lab. In Finney *et al*, published in 2023 (PMID: 37842651), we used IFN γ -reporter mice to examine which cells produce IFN γ at different time points after IAV infection. In one set of experiments, detailed in Fig 6, we compared IFN γ + cells in naïve, primary memory and re-infected mice. The primary and re-infected animals were examined on day 3 after the first or second infection respectively.

We examined multiple different cell types: NK, ILCs, NKT, gamma-delta T cell, ILCs and conventional CD4 and CD8 T cells, including antigen specific T cells identified using MHCI and MHCII tetramers. We have included additional text in the current manuscript to put our comment about T cells in more context:

“These data suggest that IFN γ producing immune cells act on infected cells that in turn upregulate molecules that enhance communication between immune and infected structural cells. Our previous study tracking in vivo IFN γ production using reporter mice, demonstrated an increase in IFN γ + NK cells and ILCs at day 3 following re-infection⁵³. In terms of number, conventional CD4 and CD8 T cells made up the largest population of IFN γ + cells within the lung in re-infected animals, aligning with findings that IFN γ producing CD4 and CD8 T cells accelerate virus control in mouse re-infection studies^{21,54,55} and are associated with protection in humans^{20,56-58}.”

As part of the dataset shown in original SFig 10 (previous SFig9), we examined a number of different populations in the lung at naïve, memory, primary day 2, and re-infection day 2 to determine whether they contained IAV-Nucleoprotein. These populations include different myeloid cells including alveolar and other macrophages, neutrophils, monocytes, DCs, and also B cells. The naïve and memory data from one of the datasets are included along with other data in our paper, Hargrave *et al*, Mucosal Immunology 2024. Part of this study is a detailed characterisation of different MHCII+ CD45+ cells in the lung after IAV infection. The day 2 primary and re-infection data are not in the previous study.

The numbers of these different populations are not strikingly different between day 2 primary and re-infection mice, any differences are present within the memory mice, for example, the higher level of cDC1 following IAV infection that is included in Hargrave *et al*, 2024. We also compared MFI of MHCII in these populations. The only difference we found was more MHCII on alveolar macrophages in memory and day 2 re-infection compared with naïve/day 2 primary, consistent with Li *et al* (PMID: 35776902). As we have not divided the alveolar macrophages into yolk sac versus monocyte derived cells, we don't think our data are useful to report in comparison to the data already published that characterise in much more depth how alveolar macrophages change following IAV infection.

In conclusion, we have examined multiple other cell types comparing IAV primary and re-infection. Some of these data are published (Finney *et al*). The interesting findings from the CD45+MHCII+ datasets have also been published (naïve versus memory in Hargrave *et al*). We do not think it will be useful to include the negative data on these subsets in a manuscript with a focus on lung structural cells and their interactions with T cells.

Reviewer 3: Comment 7

In Fig. 7C, they conclude that the slightly lower expression of CD25 and CD69 correlates with a memory phenotype of stromal cells, suggesting that these cells are better at presenting antigens. However, to sustain the claim of antigen presentation by these cells, it would be beneficial to analyze the proliferation of HA-TCR-specific CD4 and CD8 T cells after co-culturing with influenza-infected stromal cells.

Reviewer 3: Response 7

(see also Reviewer 1, Response 3)

In the study presented in the original manuscript, we saw **increased** CD25 and CD69 in co-cultures between infected lung CD45 negative cells and T cells isolated from the spleens of infected mice compared to control 'no-virus' cultures. We think the 'slightly lower' CD25 and CD69 expression the reviewer refers to is due to the reduction in virus levels in cultures with lung CD45 negative cells from IAV memory mice, further explored in Fig 8D (previous Fig 7D).

Reviewer 1 asked a similar question about antigen presentation and we conducted a similar experiment as the Reviewer 3 suggested using IAV-PR8 viruses that express Ovalbumin epitopes and OTI and OTII TCR Tg cells.

In this experiment, we infected lung CD45 negative cells from either naïve mice or mice infected 73 days earlier with IAV-X31 with OVA viruses or control wild-type PR8 *in vitro*. The OTI and OTII T cells were added after 24 hours and T cell activation examined after a further three days.

These data show clear CD25 and CD69 expression in co-cultures with the relevant antigen, e.g. OTII cells only upregulate these markers in co-cultures infected with PR8 expressing OVA323-339. We did see a significant but very slight increase in CD69 expression by OTI T cells in co-cultures with PR8 expressing OVA323-339. This minor upregulation of CD69 may be driven by the activation of the OTII cells in the same co-culture, as this was not observed in the co-culture with PR8 infected cells.

These data are included as new Supplementary Figure 14 and together Fig 8C (previous Fig7C), show that the CD45 negative cells can present antigen to T cells.

Reviewer 3: Minor Comment 1:

In SFig 3, the correlation between RNA-seq and nCounter analyses omits data for endothelial cells.

Reviewer 3: Minor Response 1:

Yes, Reviewer 3 is correct. We did not perform the correlation data analysis on endothelial cells. In the manuscript we state that:

‘BECs had the lowest number of genes that remained upregulated (43 genes), suggesting these cells may be more likely to return to homeostasis than epithelial cells or fibroblasts’.

We also faced technical challenges with the FACS sorted BECs, despite being present in large numbers in the digested tissue, the amount of RNA we were able to extract was low, likely due cell death. Given the low number of DEG and this technical challenge, we decided not to pursue these cells.

To clarify the text we have replaced ‘cells’ to read ‘epithelial cells and fibroblasts’ in the text:

*“To validate the analysis, we isolated RNA from lung **epithelial cells and fibroblasts** in an independent experiment from naïve mice and animals infected 30 days earlier and compared gene expression in cells from naïve and infected animals by microarray (nCounter Nanostring mouse immunology panel with 29 custom genes, Extended data 2).”*

Reviewer 3: Minor Comment 2:

In SFig 7, they utilized flow cytometry to improve the resolution of SpiB+ cells. However, they could have employed immunofluorescence of SpiB-mCherry+ lungs, which would also provide spatial co-localization of inflamed EpCAM+ and SpiB+ cells.

Reviewer 3: Minor Response 2:

Thank you to the reviewers for this suggestion. We are starting to look at the SpiB reporter lungs by immunofluorescence. The expression of SpiB in alveolar epithelial cells is clear. However, the autofluorescence of epithelial cells in bronchioles, where we would expect to find club and ciliated cells, makes it difficult to clearly identify SpiB+ cells above background. Currently, the methodology and data are not robust enough to include in a published article.

Reviewer 3: Minor Comment 3:

Fig. 5A lacks a control group showing the number of EpCAM+ NP cells in WSN-infected animals.

Reviewer 3: Minor Response 3:

Fig. 5A has been updated and now includes an additional FACS plot showing EpCAM1+ NP epithelial cells (0.002%) in WSN infected animals at the memory timepoint. The level of staining is not above the negative naïve control.

Reviewer 3: Minor Comment 4

In Fig. 5D, they claim that fibroblasts are less susceptible to infection, as reported by Boyd et al. in Nature 2020. However, they should provide the number of fibroblast NP+ cells to support this claim.

Reviewer 3: Minor Response 4:

The interferon responsive fibroblast (IRF) subset was first identified by Boyd *et al.*, in 2020. We show in Fig. 5D, reduced proportions and frequencies (percentages and absolute numbers) of IRF in re-infected mice compared to primary infection. We state that:

'While fibroblasts are much less likely to be infected, they are susceptible to local microenvironmental changes.'

To improve the clarity of the manuscript, we have re-written this sentence, and it now reads as follows:

'While fibroblasts are much less likely to be infected than epithelial cells, they are susceptible to local microenvironmental changes.'

We have included representative FACS plots showing the percentage of CD140a+ fibroblasts that are IAV-NP+ as new SFig 9B. As there are so few IAV-NP+ events (primary group only) it is not possible to generate cell number data.

Reviewer 4: Remarks to authors

Response:

We thank Reviewer 4 for contributing to the review of our manuscript.

Rebuttal 2:

Reviewers 1 and 3 were satisfied with the responses in the revised manuscript and were positive regarding the potential publication of our study. However, reviewer 2 continued to have concerns.

Review 2: Comment 1

Unfortunately, the authors did not adequately address my critical concerns in the revised manuscript. For instance, epithelial cell types are challenging to identify accurately by flow cytometry. This raised concerns about the validity of the new data presented in Figure 4.

Flow cytometry is a powerful tool to generate consistent quantitative data. We have built on published data and panels from a number of groups. We have now included a new paragraph in the manuscript to explain our panel and gating in detail and note that the full gating panel was included from the initial submission in a supplementary figure:

“We incorporated exclusion markers (CD45, CD31 CD140a) for immune, endothelial and stromal cells and gated based on a pan-epithelial marker EpCAM (CD326). Our gating strategy was further refined using additional surface markers (e.g., Sca-1, Pdpn, MHC-II, and CD24) to distinguish airway (club, ciliated, progenitor) and AT I and II epithelial subtypes. We used CD24 and Sca-1 to identify distinct airway epithelial cell populations based on investigations by McQualter et al⁴², Chen et al⁴³ and Lee et al⁴⁴. These groups identified multiple progenitor populations in the upper airway, CD24⁻Sca-1^{+42,43} and CD24⁺Sca1⁻⁴². We also gated on CD24⁻Sca-1⁻ to identify alveolar epithelial cells, an approach consistent with Liang et al⁴⁵ who used this strategy to identify alveolar epithelial cells. AT I epithelial cells were distinguished from AT II based on expression of cell surface marker Pdpn and MHCII^{44,46}.”

Review 2: Comment 2

Additionally, the authors' argument regarding the role of memory T cells in protection against infection appears unconvincing. Numerous studies have already demonstrated the essential role of memory T cells, particularly tissue-resident memory T cells, in limiting early viral replication, including during influenza virus infection. The claim that “most studies examine viral titres at day 5 or 6 after re-infection” does not accurately reflect the current literature. Without compelling evidence to challenge the prevailing understanding, the rigor and robustness of the data presented in the manuscript remain questionable.

We searched for relevant papers in three ways: a PubMed search, a Google search, and checking references in reviews on this topic. We focussed on papers that have tested lung viral titres after depletion of CD4 and/or CD8 T cells during an IAV re-challenge with a different IAV strain from that used in the first infection. The earliest we found other researchers had tested viral load in re-infection was day 3. Our experiment examined day 2 and, as described by Epstein et al (PMID: 9013963), we found no difference in viral titres in control mice and those depleted of CD4 and CD8 cells.

As stated in the previous rebuttal, the reduction in virus titres found in mice 2 days following re-infection could be due to changes to a number of cell types. The in vitro study in Fig 8 demonstrates that epithelial cells are one cell type that can contribute to this early enhanced control.

We would also like to re-state that in the discussion that we agree that T cells are likely required for full viral clearance and that this statement was included in previous versions.

We have amended the text in the discussion to describe in more detail, with references, studies by other researchers asking about the role of T cells in IAV re-infection (new text highlighted in yellow):

“Both CD4 and CD8 T cells protect mice and humans from IAV re-infection, especially when priming and re-challenge strains express different hemagglutinin and neuraminidase proteins^{18,20,21,29,30,57,58}. Studies in mice that have demonstrated a requirement for CD4 and/or CD8 T cells in heterosubtypic infection have mainly examined viral titres at day 5-7 following re-infection^{21,30,91-93}. Slütter et al¹⁸ and Guo et al⁹⁴ demonstrated a role for CD8 T cells as early as day 3 post re-infection. However, both our data and that from Epstein et al²⁹ show that at the earlier time point of day 2, T cells are not required for a reduction in viral titres.

Virus is still present at this time point and protection from re-infection is enhanced by T cells^{21,29,94-97}. We found increased expression of inflammatory chemokines and molecules involved in the processing and presentation of antigen following re-infection by lung structural cells. These changes may lead to an enhanced ability to attract and communicate with local T cells. In summary, therefore, our data suggests that lung epithelial cells may play both early T cell independent and then later T cell dependent roles in accelerating viral clearance following re-infection.”

Finally, in terms of the robustness of our data: both the in vivo and in vitro experiments were done twice in independent experiments with relevant controls and the plaque assay counts were carried out by a blinded observer. We refute the suggestion that the data are lack rigor or are not robust.